# A calcium/cAMP signaling loop at the ORAI1 mouth drives channel inactivation to shape NFAT induction

Xuexin Zhang[1], Trayambak Pathak[1], Ryan Yoast[1], Scott Emrich[1], Ping Xin[1], Robert M. Nwokonko[1], Martin Johnson[1], Shilan Wu[2], Céline Delierneux[1], Maxime Gueguinou[1], Nadine Hempel[3], James W. Putney Jr.[2], Donald L. Gill [1] & Mohamed Trebak[1]

ORAI1 constitutes the store-operated $Ca^{2+}$ release-activated $Ca^{2+}$ (CRAC) channel crucial for life. Whereas ORAI1 activation by $Ca^{2+}$-sensing STIM proteins is known, still obscure is how ORAI1 is turned off through $Ca^{2+}$-dependent inactivation (CDI), protecting against $Ca^{2+}$ toxicity. Here we identify a spatially-restricted $Ca^{2+}$/cAMP signaling crosstalk critical for mediating CDI. Binding of $Ca^{2+}$-activated adenylyl cyclase 8 (AC8) to the N-terminus of ORAI1 positions AC8 near the mouth of ORAI1 for sensing $Ca^{2+}$. $Ca^{2+}$ permeating ORAI1 activates AC8 to generate cAMP and activate PKA. PKA, positioned by AKAP79 near ORAI1, phosphorylates serine-34 in ORAI1 pore extension to induce CDI whereas recruitment of the phosphatase calcineurin antagonizes the effect of PKA. Notably, CDI shapes ORAI1 cytosolic $Ca^{2+}$ signature to determine the isoform and degree of NFAT activation. Thus, we uncover a mechanism of ORAI1 inactivation, and reveal a hitherto unappreciated role for inactivation in shaping cellular $Ca^{2+}$ signals and NFAT activation.

[1] Department of Cellular and Molecular Physiology, The Pennsylvania State University College of Medicine, 500 University Dr, Hershey, PA 17033, USA. [2] Department of Health and Human Services, Signal Transduction Laboratory, National Institute of Environmental Health Sciences, National Institutes of Health, 111 TW Alexander Dr, Research Triangle Park, Durham, NC 27709, USA. [3] Department of Pharmacology, The Pennsylvania State University College of Medicine, 500 University Dr, Hershey, PA 17033, USA. Correspondence and requests for materials should be addressed to X.Z. (email: xzhang3@pennstatehealth.psu.edu) or to M.T. (email: mtrebak@psu.edu)

Store-operated Ca$^{2+}$ entry (SOCE) mediated by ORAI1 plasma membrane channels is the major route for receptor-regulated Ca$^{2+}$ entry in non-excitable cells, and is crucial in controlling a plethora of physiological functions[1–3]. The biophysical manifestation of SOCE is the highly Ca$^{2+}$-selective Ca$^{2+}$ release-activated Ca$^{2+}$ (CRAC) current[4]. ORAI1-mediated Ca$^{2+}$ entry is triggered by activation of phospholipase C (PLC)-coupled receptors, generation of inositol-1,4,5-trisphosphate (IP$_3$), and subsequent depletion of endoplasmic reticulum (ER) Ca$^{2+}$ stores[1,3,5]. The cellular and physiological outcomes controlled by ORAI1 are ultimately determined by net CRAC channel activity, which is shaped by the critical balance between its store-dependent activation and its Ca$^{2+}$-dependent inactivation (CDI). Mutations in ORAI1 that disrupt CRAC channel function yield loss or gain of function channels and cause a wide spectrum of disease states, including immunodeficiency and disorders of muscle, skin, and secretory tissues[6–8].

The mechanisms linking ER Ca$^{2+}$ store depletion to activation of ORAI1 channels are now well characterized[2,9]. Depletion of Ca$^{2+}$ from the ER lumen causes the ER Ca$^{2+}$ sensor protein, stromal interaction molecule 1 (STIM1), to lose Ca$^{2+}$ binding from a low-affinity luminal EF hand, and gain an extended conformation that interacts with PM lipids within ER-PM junctions where it physically binds and opens ORAI1 to mediate CRAC current[2,9]. In contrast, the molecular mechanisms of CRAC channel CDI are less clear. CRAC channel CDI develops within milliseconds with kinetics that can be fitted to a bi-exponential curve (time constants of ~5 and 50 ms)[2,10,11]. CDI is driven by an unknown Ca$^{2+}$-dependent mechanism located either on the CRAC channel complex or situated a few nanometers from the mouth of the channel, reflecting a role of a highly localized Ca$^{2+}$ microdomain at the close vicinity of the CRAC channel pore[2].

Recently, we discovered a naturally-occurring short translational variant of ORAI1, which we termed ORAI1β. ORAI1β is generated from an alternative translation-initiation site on *Orai1* mRNA at methionine-64 producing a protein lacking the first 63 N-terminal amino acids[12,13] (Fig. 1a). Here we show that while ORAI1β mediates typical CRAC current, it has greatly reduced CDI, suggesting the first 63 N-terminal residues of ORAI1 are crucial for CDI. Using this crucial knowledge, we show that the Ca$^{2+}$-sensitive adenylyl cyclase 8 (AC8)[14], which is constitutively bound to an ORAI1 N-terminal site containing three arginines (31, 32, and 33), is crucial for CDI. Generation of cyclic adenosine monophosphate (cAMP) by AC8 activates PKA to induce CDI through direct phosphorylation of serine-34 in ORAI1 in an A-kinase-anchoring protein 79 (AKAP79)-dependent manner. Recruitment of the phosphatase calcineurin antagonizes the effect of PKA and reduces CDI. Importantly, we show that CDI shapes the frequency of Ca$^{2+}$ oscillations triggered by physiological concentrations of agonizts. When reconstituted to physiological levels in ORAI1-knockout cells, ORAI1 shows significantly lower frequency of Ca$^{2+}$ oscillations compared with ORAI1β. Consequently, NFAT4, an NFAT isoform sensitive to small increases in cytosolic Ca$^{2+}$ [15] (but not NFAT1, which requires robust cytosolic Ca$^{2+}$ for its nuclear translocation[16,17]), is translocated to the nucleus slower and less efficiently in ORAI1-expressing cells compared with ORAI1β-expressing cells. Our findings identify a molecular mechanism for ORAI1 CDI and provide evidence that CDI, driven by spatially-restricted Ca$^{2+}$-cAMP crosstalk, plays a crucial role in shaping cellular signaling and NFAT activation.

## Results

**ORAI1 displays greater CDI compared with ORAI1β.** Through alternative translation-initiation, *Orai1* mRNA produces ORAI1

protein from methionine-1 and a shorter ORAI1β protein starting at methionine-64[12,13] (Fig. 1a). These native ORAI1 isoforms can be resolved from HEK293 cells as two distinct bands in western blots, but only if protein samples are deglycosylated by PNGase F (Supplementary Fig. 1a, b; compare lane 1 with 5). To uncover potential differences between these two isoforms, we used CRISPR/Cas9 technology to generate ORAI1-knockout HEK293 (ORAI1-KO) cells, which offered a clean background in which each isoform could be individually expressed. ORAI1 knockout was confirmed by genomic sequencing, western blot (Supplementary Fig. 1a, b; compare lane 1 with 2 and lane 5 with 6; also see Supplementary Fig. 1c), and Fura-2 Ca$^{2+}$ imaging of SOCE triggered by store depletion using the sarcoplasmic/endoplasmic reticulum Ca$^{2+}$ ATPase (SERCA) blocker, thapsigargin at 2 µM (Supplementary Fig. 1d, e).

ORAIs C-terminally tagged with CFP were engineered to exclusively express either ORAI1 or ORAI1β (see Methods). The expression of ORAI1-CFP and ORAI1β-CFP were driven by either the strong cytomegalovirus (CMV) promoter (Supplementary Fig. 1b; to ensure big reliable currents in patch clamp) or the weak thymidine kinase (TK) promoter (Supplementary Fig. 1a; to reconstitute physiological levels of ORAI1 expression for functional studies). When ORAI1-CFP and ORAI1β-CFP were independently co-expressed with eYFP-STIM1 in ORAI1-KO cells, both isoforms generated CRAC currents with identical properties. However, when CDI was measured with 10 mM EGTA in the patch pipette using a voltage-step protocol (Supplementary Fig. 1f; see Methods), CRAC currents mediated by ORAI1 showed robust CDI compared with those mediated by ORAI1β (Fig. 1b, c). The remaining current mediated by ORAI1 at 146 ms was smaller compared with ORAI1β (Fig. 1d) and both time constants of inactivation (see Methods) were greater for ORAI1 compared with ORAI1β (Supplementary Fig. 1g, h). The difference in CDI between ORAI1 and ORAI1β, which was highly significant when CRAC currents were activated by including 10 mM of the slow buffer EGTA in the pipette (Fig. 1b–d) disappeared when CRAC currents were activated by dialysis of 20 mM of the fast chelator BAPTA through the pipette (Fig. 1e–g) or when extracellular Ca$^{2+}$ was replaced by Ba$^{2+}$ (Fig. 1h–j), strongly arguing that this difference in CDI is mediated by cytosolic Ca$^{2+}$ at the vicinity of ORAI1 channels. Supplementary Table 4 contains a summary of patch clamp data reported throughout the study with statistical analysis. The finding that these two naturally-occurring ORAI1 isoforms have distinct CDI properties argued that the 63 additional residues of ORAI1 are responsible for this difference in CDI, and provided a unique opportunity to unravel mechanistic details underlying ORAI1 CDI.

**AC8 and caveolin-binding sites of ORAI1 are required for CDI.** We focused our attention on the 63 residues unique to ORAI1 (Fig. 1a). Instead of performing serial truncations of this 63 amino-acid region, we mutated specific residues within domains with putative functions, including residues potentially phosphorylated or mediating interactions with other molecules[3]. Within these 63 amino acids, there are two putative protein kinase C (PKC) phosphorylation sites (Serine-27 and Serine-30)[18,19], an AC8-binding domain (residues 26–34)[14] that overlaps with a putative phosphatidylinositol-4,5-bisphosphate (PIP$_2$)-binding sequence (residues 28–33)[20], and a caveolin (cav) binding domain (residues 52–60)[21] (see Fig. 1a).

To test the hypothesis that Ca$^{2+}$-mediated generation of cAMP by AC8 at the vicinity of the ORAI1 pore is mediating CDI, we generated two different constructs: (1) an ORAI1-CFP construct deficient in AC8 binding in which arginines 31–33 were mutated

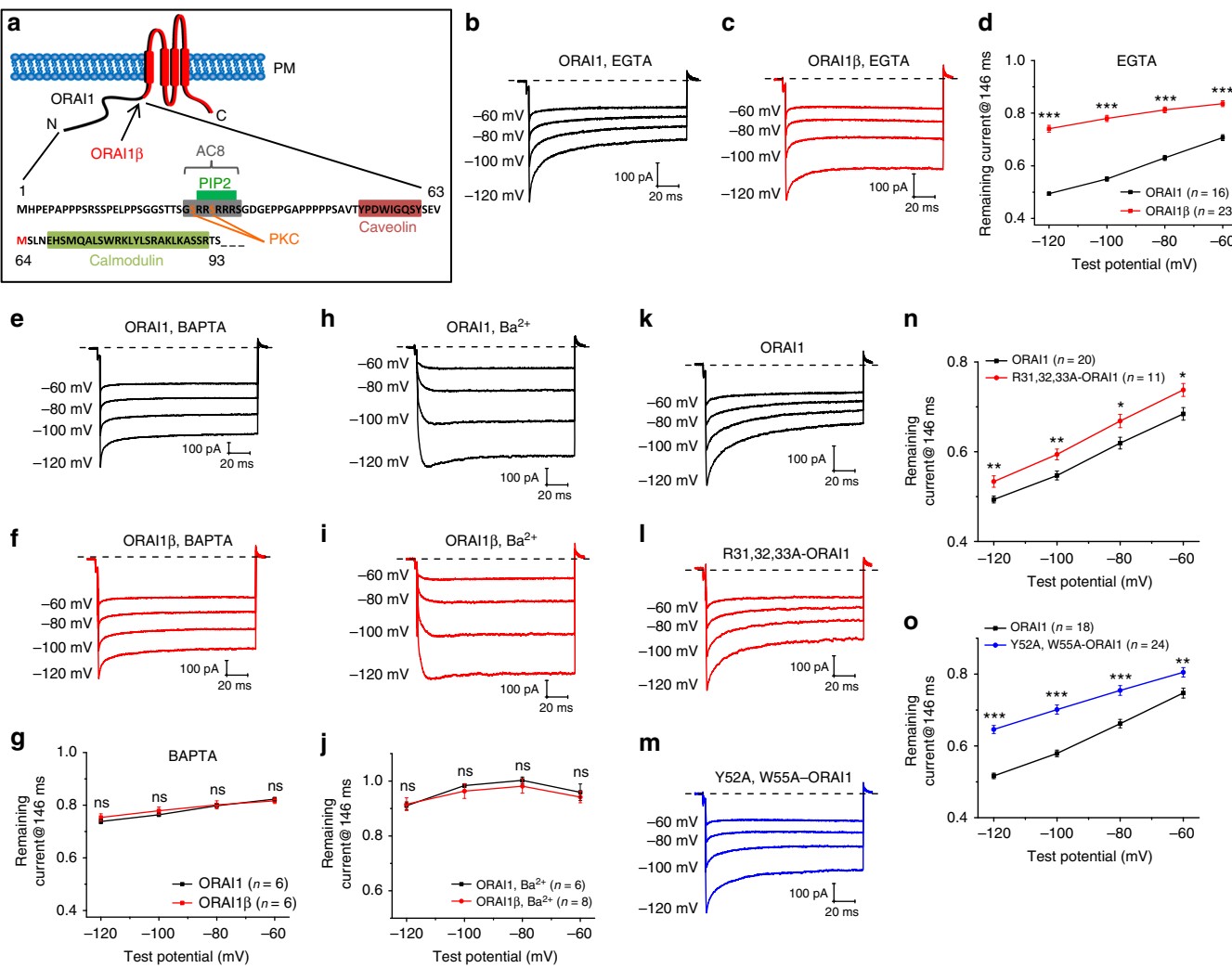

**Fig. 1** AC8- and cav-binding site on ORAI1 are required for ORAI1 CDI. **a** Schematic of consensus domains unique to ORAI1. The first N-terminal 63 amino acids are exclusive to ORAI1 (black). ORAI1β (red) starts at Methionine-64. The putative calmodulin (CaM)-binding domain is found in both ORAI1 and ORAI1β. Recordings are from ORAI-KO cells co-expressing eYFP-STIM1 with either (**b**, **e**, **h**) ORAI1-CFP or (**c**, **f**, **i**) ORAI1β-CFP. Representative currents using either 10 mM EGTA (**b**, **c**), or 20 mM BAPTA (**e**, **f**) in patch pipette with 20 mM $Ca^{2+}$ bath solution, or 10 mM EGTA in patch pipette with 20 mM $Ba^{2+}$ bath solution (**h**, **i**). CDI was revealed by applying a voltage-step protocol from a holding potential of + 30 mV as depicted in (Supplementary Fig. 1f; see Methods). **d**, **g**, **j** The extent of CRAC channel inactivation for ORAI1 and ORAI1β represented as current remaining at the end of the pulse (at 146 ms; see Methods). Each data point represents mean ± SEM. **k–o** Representative currents from ORAI-KO cells co-expressing eYFP-STIM1 with either **k** WT ORAI1-CFP, **l** ORAI1-CFP mutant deficient in AC8-binding (R31–33A), or **m** ORAI1-CFP mutant deficient in caveolin binding (Y52A, W55A). The extent of CDI for ORAI1 R31–33A (**n**) and ORAI1 Y52A, W55A (**o**) compared with the respective side-by-side recordings from WT ORAI1 are represented as mean ± SEM. *$p < 0.05$, **$p < 0.01$, ***$p < 0.001$, ns, not significant, two-tailed Student's $t$ test was used for (**d**, **g**, **j**, **n**, **o**)

to alanines (R31–33A) and (2) because adenylyl cyclase enzymes are typically located within cav-rich regions of the plasma membrane[22], we generated an ORAI1-CFP construct deficient in cav binding in which residues tyrosine-52 and tryptophane-55 were mutated to alanines (Y52A, W55A). Co-expression of either construct with eYFP-STIM1 in ORAI1-KO cells revealed that both mutants have significantly reduced CDI compared with wild-type ORAI1 (Fig. 1k–o), suggesting a role for AC8 binding to ORAI1 in mediating CDI. Although statistically significant, this difference in CDI between wild-type ORAI1 and R31–33A and Y52A, W55A ORAI1 mutants (Fig. 1k–o) was less than the difference in CDI between ORAI1 and ORAI1β (Fig. 1b–d), suggesting that ORAI1 CDI involves other mechanisms in addition to AC8.

We ruled out a role for PKC-mediated phosphorylation of ORAI1 by showing that the extent of CDI of an ORAI1-CFP construct in which the PKC phosphorylation sites serine-27 and

serine-30 were mutated to alanine (S27A, S30A), was similar to wild-type ORAI1-CFP (Supplementary Fig. 1i–k). Furthermore, ORAI1-KO cells expressing wild-type ORAI1-CFP and treated with a specific PKC inhibitor (Gö6983; 10 μM) showed similar CDI to cells treated with vehicle control (Supplementary Fig. 1l–n). Dialysis of diC8-PIP₂ through the patch pipette at concentrations as high as 30 μM ruled out a role for PIP₂ in ORAI1 CDI (Supplementary Fig. 2a–c). Therefore, we focused our attention on the molecular mechanisms of AC8 regulation of ORAI1 CDI.

**AC8/ORAI1 interaction and cAMP are required for ORAI1 CDI.** To rule out spurious effects of mutagenesis on ORAI1 CDI, we determined the effect of AC8 protein knockdown and over-expression on CDI of CRAC currents generated by co-expression of eYFP-STIM1 and ORAI1-CFP. Protein knockdown of AC8 by

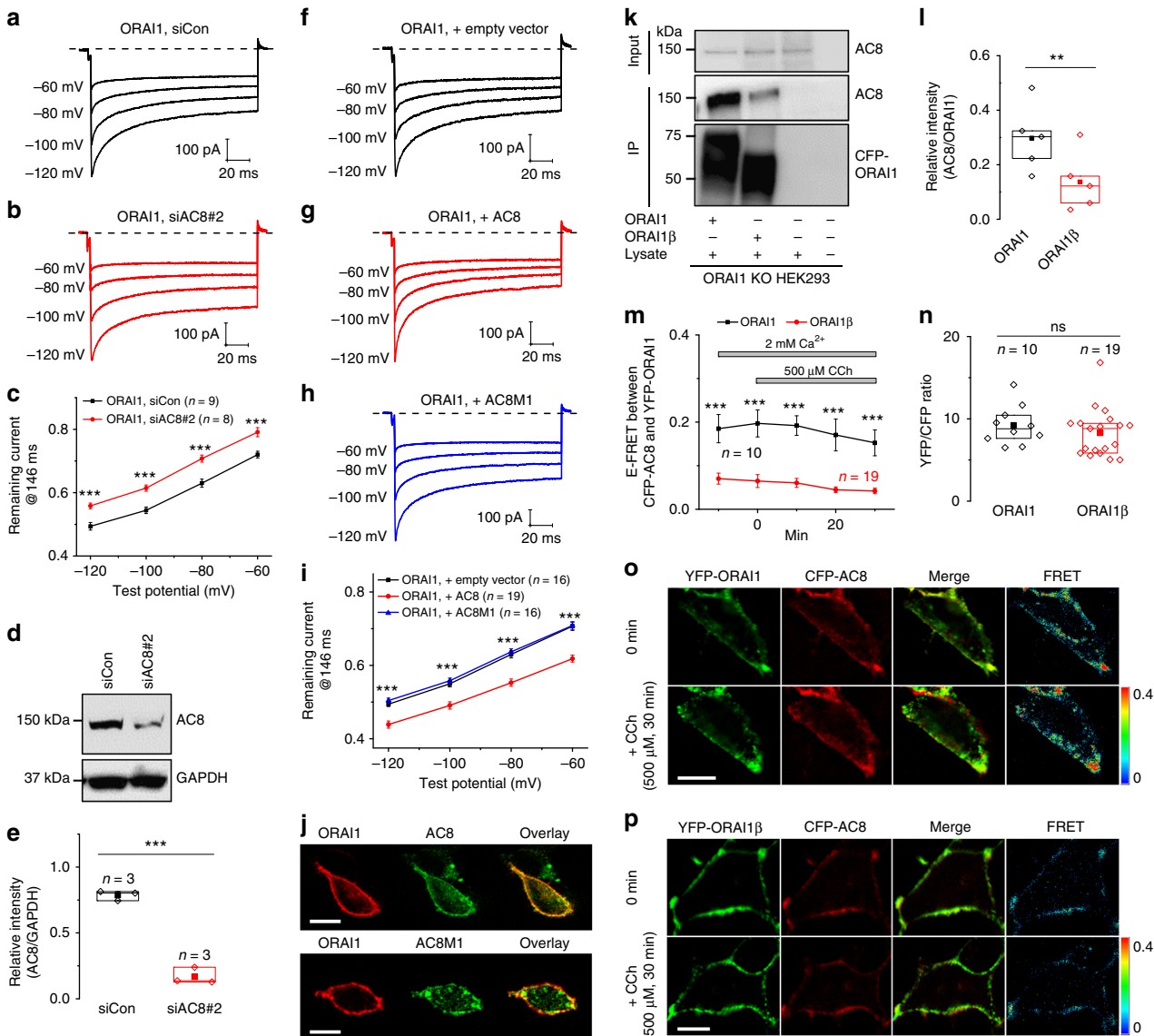

**Fig. 2** AC8 mediates ORAI1 CDI through physical interaction with ORAI1. **a–c** Representative currents from ORAI1-KO cells co-expressing eYFP-STIM1 with ORAI1-CFP and co-transfected with **a** control non-targeting siRNA or **b** siRNA against AC8. **c** The extent of ORAI1 CDI from (**a**) and (**b**) represented as current remaining at 146 ms. Each data point represents mean ± SEM. **d** Western blot of AC8 in cells transfected with control non-targeting siRNA and AC8 siRNA. **e** Densitometry of AC8/GAPDH proteins from three independent experiments and shown as boxplots representing the 25th to 75th percentile range, mean, and median. **f–i** Representative currents from ORAI1-KO cells co-expressing eYFP-STIM1 with ORAI1-CFP and co-transfected with **f** an empty vector control, **g** AC8 cDNA, or **h** AC8M1 cDNA. **i** The extent of ORAI1 CDI from (**f**), (**g**), and (**h**) represented as current remaining at 146 ms. Each data point represents mean ± SEM. **j** Fluorescence image of ORAI1-KO cells co-expressing ORAI1-CFP (red) and eYFP-tagged versions (green) of either AC8 (top) or AC8M1 (bottom). Scale bar: 10 μm (**k**, **l**). **k** Pull-down assay of AC8 with ORAI1-CFP and ORAI1β-CFP expressed in ORAI1-KO cells. **l** Quantification of AC8 pull-down data from five independent experiments and represented as boxplots showing the 25th to 75th percentile range, mean, and median. **m** e-FRET data of interactions between CFP-AC8 and either eYFP- ORAI1 or eYFP-ORAI1β before and after addition of carbachol (Cch, 500 μM) with each point representing mean ± SEM. **n** YFP/CFP fluorescence ratios from recordings in (**m**) represented as boxplots showing the 25th to 75th percentile range, mean, and median. **o**, **p** Representative fluorescence and FRET images of cells expressing **o** CFP-AC8 and eYFP- ORAI1, or **p** CFP-AC8 and eYFP-ORAI1β. Scale bar: 10 μm. *$p < 0.05$, **$p < 0.01$, ***$p < 0.001$, ns, not significant, two-tailed Student's $t$ test was used for (**c**, **e**, **l**, **m**, **n**), and one-way ANOVA for (**i**)

~65% (Fig. 2d, e) significantly inhibited ORAI1 CDI (Fig. 2a–c). AC8 protein knockdown had no additional effect on CDI of the R31-33A-ORAI1-CFP mutant that is deficient in AC8 binding (Supplementary Fig. 2d–f), supporting a role for AC8-ORAI1 interactions in mediating ORAI1 CDI. Importantly, AC8 knockdown had no effect on ORAI1β CDI (Supplementary Fig. 2g–i), while side by side recordings showed that ORAI1 CDI is inhibited (Supplementary Fig. 2j–l), suggesting that close and specific association of AC8 with ORAI1 is a crucial determinant for CDI. Conversely, overexpression of AC8 cDNA (Supplementary Fig. 2m) enhanced ORAI1 CDI (Fig. 2f–i). However, overexpression of an AC8 N-terminal truncation mutant, which lacks the first 106 residues and is deficient in ORAI1 binding[14] (AC8M1; Fig. 2j) failed to affect ORAI1 CDI (Fig. 2h, i). The Pearson's coefficient of co-localized volume between ORAI1/AC8 is 0.712 ± 0.034 versus 0.4675 ± 0.0588 for ORAI1/AC8M1 ($n = 5$–8; Fig. 2j); also see FRET data in Willoughby et al.[14].

Pull-down assays showed that AC8 interacts robustly with ORAI1-CFP but not with ORAI1β-CFP (Fig. 2k, l). Förster

resonance energy transfer (FRET) microscopy (Fig. 2m–p) further supported these results. For these FRET experiments, constructs of AC8, ORAI1, and ORAI1β were tagged on their N-termini. Unlike eYFP-ORAI1β, eYFP-ORAI1 showed a robust constitutive interaction with CFP-AC8 (Fig. 2m–p), which was not altered after maximal agonist stimulation (500 μM carbachol; Fig. 2m–p).

The next logical step was to determine the role of AC8 enzymatic product, cAMP in CDI. We determined whether enhancing cellular cAMP levels through either activation of adenylyl cyclase by forskolin or by direct dialysis of cAMP through the pipette would have an effect on ORAI1 CDI. Forskolin addition to the bath solution (10 μM; Fig. 3a–c) or dialysis of cAMP (100 μM; Fig. 3d–f) through the pipette, in both cases significantly enhanced CDI. However, addition of forskolin to the bath (Supplementary Fig. 3a-c) or cAMP to the pipette

(Supplementary Fig. 3d-f) at the same concentrations failed to induce CDI in ORAI1β, indicating that a global cytosolic increase in cAMP is not sufficient to rescue ORAI1β CDI and that cAMP action requires the 63 amino acids exclusive to ORAI1.

**PKA interaction with AKAP79 are required for ORAI1 CDI.** The major cytosolic targets for cAMP are protein kinase A (PKA) and the exchange factor directly activated by cAMP (EPAC)[23]. Addition of a membrane-permeant and specific activator of EPAC proteins, 8-pCPT-AM at 10 μM to the bath, had no effect on CDI of ORAI1 (Supplementary Fig. 4a–c), ruling out EPAC as the mediator of cAMP action on ORAI1 inactivation. However, a specific inhibitor of PKA, H89 (10 μM) inhibited ORAI1 CDI to the same level whether CDI was recorded under basal conditions (Fig. 3g–i) or when enhanced by inclusion of 100 μM cAMP

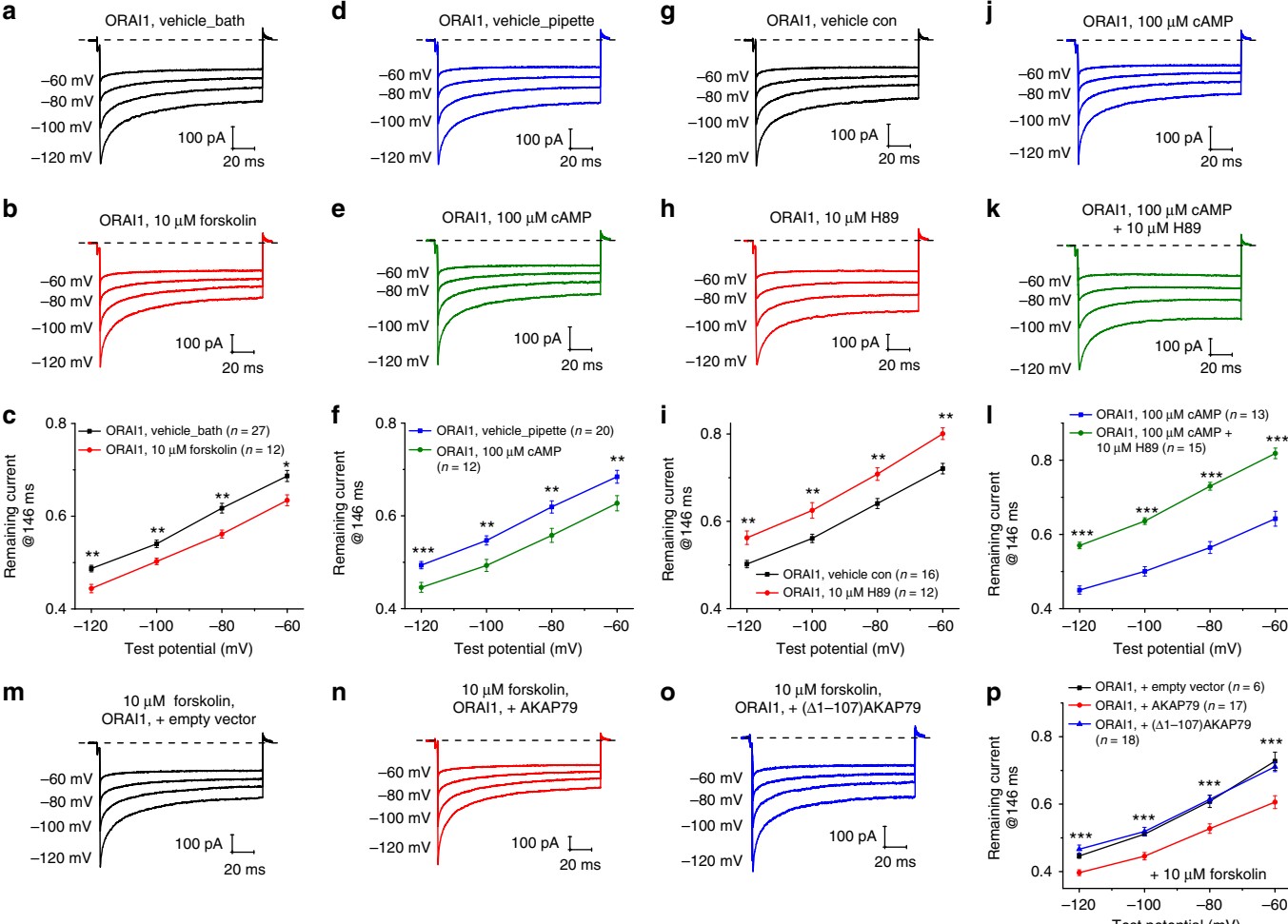

**Fig. 3** ORAI1 CDI requires cAMP and PKA. **a–c** Representative currents from ORAI1-KO cells co-expressing eYFP-STIM1 with ORAI1-CFP and treated in the bath with either **a** vehicle or **b** Forskolin (10 μM). **c** The extent of ORAI1 CDI from (**a**) and (**b**) represented as current remaining at 146 ms. Each data point represents mean ± SEM. **d–f** Representative currents from ORAI1-KO cells co-expressing eYFP-STIM1 with ORAI1-CFP and dialyzed through patch pipette with either **d** vehicle or **e** cAMP (100 μM). **f** The extent of ORAI1 CDI from (**d**) and (**e**) represented as current remaining at 146 ms. Each data point represents mean ± SEM. **g–i** Representative currents from ORAI1-KO cells co-expressing eYFP-STIM1 with ORAI1-CFP and treated in the bath with either **g** vehicle or **h** PKA inhibitor (10 μM). **i** The extent of ORAI1 CDI from (**g**) and (**h**) represented as current remaining at 146 ms. Each point represents mean ± SEM. **j–l** Representative currents from ORAI1-KO cells co-expressing eYFP-STIM1 with ORAI1-CFP dialyzed with cAMP (100 μM) through the patch pipette and treated in the bath with either **j** vehicle or **k** PKA inhibitor (10 μM). **l** The extent of ORAI1 CDI from (**j**) and (**k**) represented as current remaining at 146 ms. Each data point represents mean ± SEM. **m–p** Representative currents from ORAI1-KO cells co-expressing eYFP-STIM1 with ORAI1-CFP stimulated with Forskolin (10 μM) and expressing either **m** an empty vector, **n** full-length AKAP79, or **o** Δ1-107AKAP79. **p** The extent of ORAI1 CDI from (**m**), (**n**), and (**o**) represented as current remaining at 146 ms. Each data point represents mean ± SEM. *$p < 0.05$, **$p < 0.01$, ***$p < 0.001$, two-tailed Student's $t$ test was used for (**c**, **f**, **i**, **l**), and one-way ANOVA for (**p**)

(Fig. 3j–l), suggesting a role of PKA in ORAI1 CDI. Similar results were obtained with the use of a different PKA inhibitor, KT5720 (5 μM; Supplementary Fig. 4d–f) while the same concentration of a structurally-related protein kinase G (PKG) inhibitor, KT5823 (5 μM) was without effect on ORAI1 CDI (Supplementary Fig. 4g–i).

AKAP proteins are essential for PKA function; they bind PKA regulatory subunits and confine PKA to discrete locations near its specific substrates[24]. Therefore, we set out to determine whether the AKAP isoform widely expressed in HEK293 cells, AKAP79 is preferentially associated with ORAI1 and required for ORAI1 CDI. We used two independent conditions where we expressed either full-length AKAP79 cDNA or a truncated version of AKAP79 lacking the first 107 residues (Δ1-107AKAP79), corresponding to the membrane-targeting domain[25]. We expected that only full-length AKAP79 would mediate enhancement of ORAI1 CDI. However, ORAI1 CDI was unaffected by expression of either AKAP79 or Δ1-107AKAP79 (Supplementary Fig. 5a–d), suggesting that, as previously reported[26], under basal conditions endogenous AKAP79 in HEK293 cells is not limiting. Indeed, when ORAI1 CDI was enhanced by inclusion of forskolin (10 μM) in the bath, expression of AKAP79 further enhanced ORAI1 CDI while Δ1-107AKAP79 was without effect (Fig. 3m–p). Moreover, we used FRET microscopy to show that AKAP79 associates with ORAI1, but not with ORAI1β, and this association is not altered by supramaximal stimulation of agonist (500 μM carbachol; Supplementary Fig. 5e–h).

To provide evidence for specific interactions between PKA and AKAP79 in ORAI1 CDI, we performed recordings where we included in the pipette the stearated form of a peptide termed st-Ht-31 (DLIEEAASRIVDAVIEQVKAAGAY; 10 μM), which inhibits the interaction between the RII regulatory subunit of PKA and AKAP[27]. We also performed separate recordings with a corresponding negative control peptide (st-Ht-31P; 10 μM) with disrupted structure by substitution of two isoleucine residues with proline (DLIEEAASRPVDAVPEQVKAAGAY). ORAI1 CDI was inhibited by the st-Ht-31 peptide whereas the control st-Ht-31P peptide that is unable to inhibit PKA-AKAP79 interactions, had no effect (Fig. 4a–c).

**Serine-34 on ORAI1 is phosphorylated by PKA and mediates CDI.** We reasoned that PKA phosphorylation of ORAI1 likely mediates the effects of PKA on ORAI1 CDI. Although it is unlikely that PKA-mediated phosphorylation of STIM1 could mediate CDI exclusively in ORAI1 without affecting ORAI1β, we also considered this possibility. In silico prediction (phosphosite. org, and mbc.nctu.edu.tw) predicted serine-34 on ORAI1 (a residue missing in ORAI1β) and threonine-389 on STIM1 as strong candidates for PKA phosphorylation. PKA-mediated phosphorylation of threonine-389 on STIM1 was recently shown to mediate the activation of store-independent arachidonate-regulated Ca²⁺ (ARC) channels formed by heteromeric associations between ORAI1 and ORAI3[28]. Wild-type STIM1-pVenus, and two STIM1 mutants in which threonine-389 was mutated to either alanine (T389A STIM1-pVenus; phosphodeficient) or glutamate (T389E STIM1-pVenus; phosphomimetic) were co-expressed with ORAI1-CFP in STIM1/STIM2 double knockout HEK293 cells that we generated using CRISPR/Cas9 technology and characterized for completely lacking both STIM1 and STIM2 proteins and SOCE (Supplementary Fig. 6a–c). Both T389A and T389E STIM1 mutants were capable of supporting ORAI1 CDI that is comparable to wild-type STIM1 (Supplementary Fig. 6d–g), indicating that threonine-389 is not involved in regulating ORAI1 CDI.

We also generated several ORAI1-CFP mutants of serine-34 which we co-expressed with eYFP-STIM1 in ORAI1-KO cells. We generated a serine-34 to alanine (S34A) mutant, a mutation of serine-34 to a positively charged residue, arginine (S34R) and two phosphomimetic mutants with either glutamate (S34E) or aspartate (S34D). The CDI of the phosphomimetic mutants (S34E and S34D) was indistinguishable from that of wild-type ORAI1 (Fig. 4d, g–i). Interestingly, ORAI1 S34A consistently showed significantly reduced CDI while ORAI1 S34R showed even more reduced CDI compared with ORAI1 S34A (Fig. 4d–f, i), implicating the negative charges added through phosphorylation on serine-34 located at the extension of ORAI1 pore, in mediating ORAI1 CDI. Further, we showed that unlike wild-type ORAI1, ORAI1 S34A did not show enhancement of CDI when cAMP was included in the pipette (Supplementary Fig. 6h–k), strongly arguing that serine-34 is the target of PKA-mediated phosphorylation on ORAI1. We also showed that ORAI1 CDI is driven by Ca²⁺-mediated production of cAMP, because the difference in CDI between wild-type ORAI1 and S34A ORAI1 observed in Ca²⁺-containing bath solutions disappeared when recordings were performed in Ba²⁺-containing bath solutions (Supplementary Fig. 7a–d).

CRAC channels are the subject of a less specific slow inactivation process driven by global rise of cytosolic Ca²⁺, which occurs within tens of seconds to minutes of activation (Supplementary Fig. 7e)[29]. We show that slow inactivation of ORAI1, ORAI1β and S34R ORAI1 (co-expressed with eYFP-STIM1) are indistinguishable (Supplementary Fig. 7f, g), supporting cAMP signaling as a specific and local regulator of ORAI1 CDI.

To provide evidence that serine-34 is phosphorylated by PKA, we performed phosphorylation assays with ORAI-KO cells co-expressing STIM1 with either C-terminal eGFP-tagged ORAI1 or ORAI1-S34A. We used forskolin (10 μM) to stimulate cAMP production and ORAI1 phosphorylation was detected by western blot with an anti-phosphoserine specific antibody on immunoprecipitated ORAI1-eGFP proteins (Fig. 4j, k). These assays clearly demonstrated that ORAI1 phosphorylation is greatly enhanced in response to stimulation with forskolin. However, ORAI1 S34A showed undetectable phosphorylation either in the absence or presence of forskolin (Fig. 4j, k), strongly arguing that PKA induces phosphorylation of serine-34 on ORAI1 to mediate CDI and that serine-34 is the major target for PKA phosphorylation of ORAI1.

AC8-mediated signaling events culminating in ORAI1 phosphorylation are likely not fast enough (5–50 ms range) to serve as the actual trigger for ORAI1 CDI[30]. Comparison of S34A-ORAI1 CDI measured with either 10 mM EGTA or 20 mM BAPTA in the pipette showed that CDI of S34A-ORAI1 is significantly reduced in BAPTA compared with EGTA (Supplementary Fig. 8a–c), suggesting that a distinct intracellular Ca²⁺ binding site is linked to ORAI1 CDI and that phosphorylation of serine-34 likely contributes to CDI by recruiting more ORAI1 channels into an inactive state.

ORAI1, S34A-ORAI1, and ORAI1β showed identical levels of PM expression and peak currents. Nevertheless, bigger unitary conductance and open probability of ORAI1 compared with S34A-ORAI1 and ORAI1β might contribute to its enhanced CDI. Bigger single channel conductance might lead to higher Ca²⁺ concentrations next to individual ORAI1 channels at sites inaccessible to BAPTA (see, e.g., ref. [31]). Due to their extremely low single channel activity[32], direct measurements of CRAC single channel conductance is not possible. As an alternative, we compared patch clamp recordings performed with 10 mM EGTA in the pipette and 20 mM Ca²⁺ in bath to recordings performed with a combination of weaker intracellular buffer (1 mM EGTA

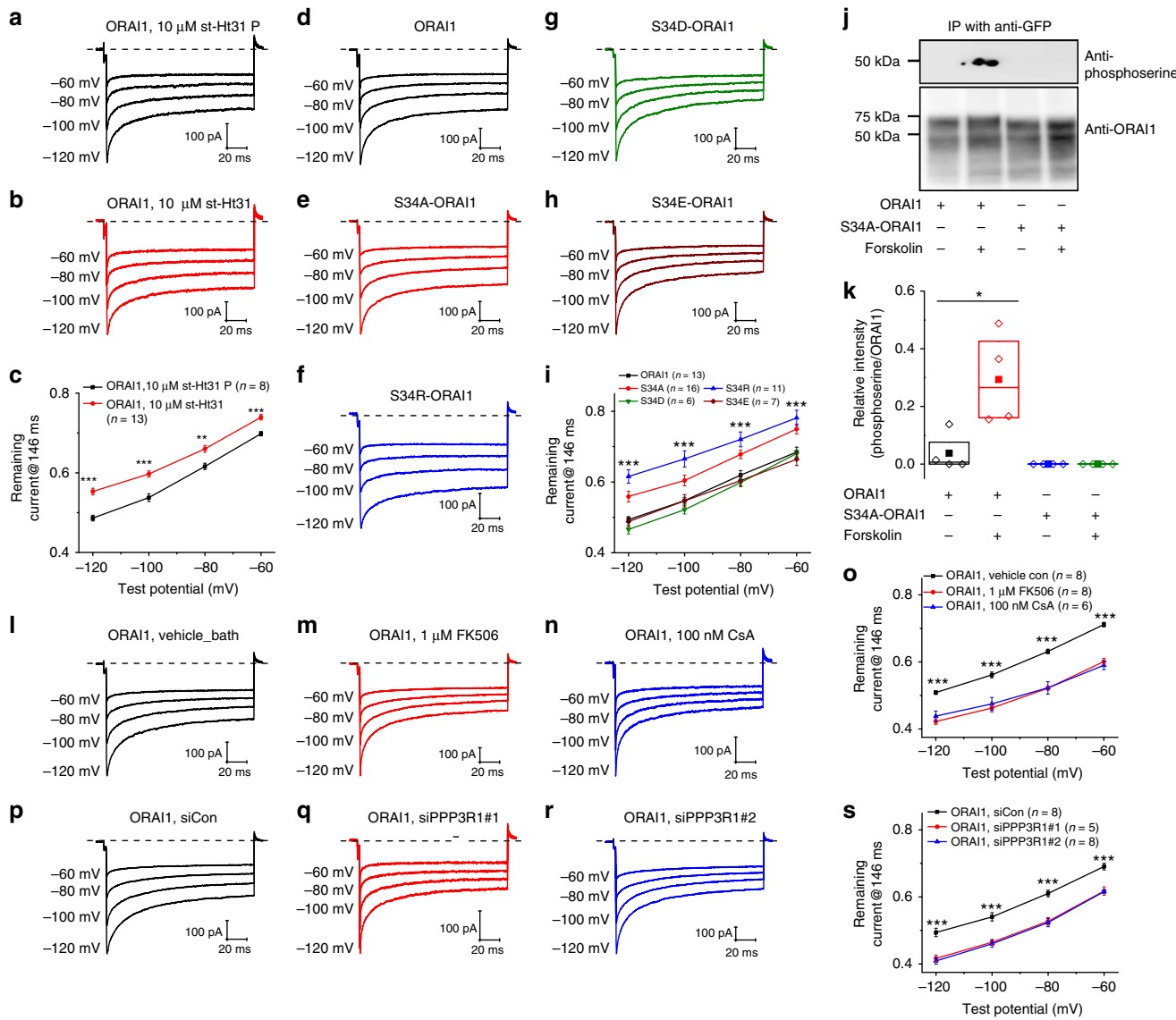

**Fig. 4** PKA/AKAP79 and calcineurin reciprocally regulate ORAI1 CDI. **a–c** Representative currents from ORAI1-KO cells co-expressing eYFP-STIM1 with ORAI1-CFP and dialyzed through patch pipette with 10 μM of either **a** the negative control st-Ht31 P peptide or **b** the st-Ht31 peptide. **c** The extent of ORAI1 CDI from (**a**) and (**b**) represented as current remaining at 146 ms. Each data point represents mean ± SEM. **d–i** Representative currents from ORAI1-KO cells co-expressing eYFP-STIM1 with either **d** wild-type ORAI1-CFP, **e** S34A ORAI1-CFP, **f** S34R ORAI1-CFP, **g** S34D ORAI1-CFP, or **h** S34E ORAI1-CFP. **i** The extent of CDI from all ORAI1 variants represented as current remaining at 146 ms. Each data point represents mean ± SEM. **j** Anti-phosphoserine western blots on immunoprecipitated wild-type ORAI1-eGFP and S34A ORAI1-eGFP from proteins extracts of ORAI1-KO cells co-expressing eYFP-STIM1 with and without stimulation with Forskolin (10 μM). **k** Densitometry of phosphoserine/ORAI1 ratios from several independent experiments are shown as boxplots representing the 25th to 75th percentile range, mean and median. **l–o** Representative currents from ORAI1-KO cells co-expressing eYFP-STIM1 with ORAI1-CFP and treated in the bath with either **l** vehicle, **m** FK506 (1 μM), or **n** cyclosporine A (CsA, 100 nM). **o** The extent of ORAI1 CDI from (**l**), (**m**), and (**n**) represented as current remaining at 146 ms. Each data point represents mean ± SEM. **p–s** Representative currents from ORAI1-KO cells co-expressing eYFP-STIM1 with ORAI1-CFP and co-transfected with either **p** a control non-targeting siRNA, or **q**, **r** two independent siRNA sequences against calcineurin regulatory subunit PPP3R1. **s** The extent of ORAI1 CDI from (**p**), (**q**), and (**r**) represented as current remaining at 146 ms. Each data point represents mean ± SEM. *$p < 0.05$, **$p < 0.01$, ***$p < 0.001$, two-tailed Student's $t$ test was used for (**c**, **k**), and one-way ANOVA for (**i**, **o**, **s**)

in pipette, and 30 μM 1,4,5-IP$_3$ to deplete the stores) and enhanced driving force (50 mM $Ca^{2+}$ in bath). We show that differences in CDI between ORAI1, S34A-ORAI1, and ORAI1β persist when a combination of low cytosolic buffer and enhanced $Ca^{2+}$ driving force is used (Supplementary Fig. 9), suggesting that potential differences in unitary conductance between these channel variants are unlikely to explain their different CDI.

STIM1 participates in CRAC channel inactivation because the extent of fast CDI correlated with the STIM1/ORAI1 transfection ratio and CRAC channels activated by the STIM1 minimal 100

amino-acid domain called STIM-ORAI activating region[33] or CRAC-activating domain[34] (SOAR/CAD; residues 342–448) showed reduced fast CDI[34–36]. Subsequent studies used truncated constructs and identified a negatively charged region of STIM1 (residues 470–491) required for fast CDI[10,11,37,38]. Although both ORAI1 and ORAI1β are gated by STIM1 and both generate robust STIM1-dependent currents, weaker STIM1 binding to ORAI1β versus ORAI1 might account for reduced ORAI1β CDI. We measured ORAI1 and ORAI1β interactions with STIM1 using FRET microscopy under basal conditions, after store depletion

with 2 μM thapsigargin and in presence of the PKA inhibitor H89 (10 μM). Unexpectedly, we discovered that on store depletion ORAI1/STIM1 interactions were significantly less than those of ORAI1β/STIM1 and PKA inhibition had no effect on those interactions (Supplementary Fig. 10), suggesting that ORAI1 and ORAI1β difference in CDI is independent of STIM1 binding.

**Calcineurin opposes PKA-mediated enhancement of ORAI1 CDI**. AKAP79 binds not only to PKA and PKC but also to the serine/threonine phosphatase, calcineurin. Calcineurin, which is known to mediate dephosphorylation that opposes PKA-mediated phosphorylation[28], interacts with AKAP79 and localizes near CRAC channels[39]. Pharmacological inhibition of calcineurin with two specific inhibitors, FK506 (1 μM) or cyclosporine A (100 nM) caused a significant enhancement of ORAI1 CDI (Fig. 4l–o), suggesting that the balance between PKA and calcineurin activities determines the extent of ORAI1 CDI. These results were substantiated with molecular knockdown of the regulatory subunit 1 of calcineurin (PPP3R1), which caused a significant enhancement of ORAI1 CDI (Fig. 4p–s). RT-qPCR experiments determined that HEK293 cells predominantly express *PPP3R1* mRNA with undetectable levels of *PPP3R2* mRNA (Supplementary Fig. 11a). We also documented that knockdown of *PPP3R1* led to a significant decrease in *PPP3R1* mRNA (Supplementary Fig. 11b) and protein (Supplementary Fig. 11c, d).

**CDI regulates Ca$^{2+}$ oscillations and NFAT4 translocation**. In response to low physiological concentrations of agonist (5–15 μM carbachol), Ca$^{2+}$ signals take the shape of regenerative Ca$^{2+}$ oscillations, the frequency of which is decoded by specific transcription factors to drive specific gene programs[40]. These oscillations are driven by Ca$^{2+}$ release from the ER via IP$_3$ receptors but are sustained with time only when Ca$^{2+}$ entry through CRAC channels is active, which is necessary to replenish ER Ca$^{2+}$ stores[41]. We sought to determine whether enhanced CDI of ORAI1, which is associated with less Ca$^{2+}$ entry through CRAC channels[13], translates into reduced oscillatory Ca$^{2+}$ signaling in ORAI1-expressing cells stimulated with low physiological concentrations of carbachol. To ascribe physiological significance to our findings in these critical experiments, we used HEK293 ORAI1-KO cells in which we introduced plasmids encoding either ORAI1-CFP or ORAI1β-CFP cDNA that were subcloned to be expressed under the control of a weak thymidine kinase (TK) promoter[42] (Supplementary Fig. 11e, f; see also Supplementary Fig. 1a). The use of TK promoter generates considerably less expression than studies described thus far using the stronger CMV promoter (see side by side comparison in Supplementary Fig. 11e, f; see also Supplementary Fig. 1a, b). This approach allowed rescue of each ORAI1 isoform expression in ORAI1-KO cells at or near physiological levels and circumvented the need to co-express eYFP-STIM1 cDNA, relying instead on endogenous STIM1 for ORAI1 isoform gating. By comparison to ORAI1β-expressing cells, ORAI1-expressing HEK293 cells showed reduced frequency of Ca$^{2+}$ oscillations, reduced number of cells displaying a sustained response in the shape of a plateau (defined as traces with an initial transient that does not return to baseline; representative trace in Supplementary Fig. 11o) and reduced amplitude of these plateaus in response to 10 μM (Supplementary Fig. 11h–j) and 15 μM carbachol (Fig. 5a–f; also Supplementary Fig. 11l, m). By contrast, the number of cells showing a single initial Ca$^{2+}$ spike upon stimulation with return to baseline and lack of additional response for the duration of recordings was enhanced in ORAI1-expressing HEK293 cells (Supplementary Fig. 11g, k). Representative trace for one spike event is shown in

Supplementary Fig. 11n while representative trace for regenerative oscillations shown in Supplementary Fig. 11p. Significantly, ORAI1-KO cells expressing phosphorylation-deficient S34A ORAI1-CFP and S34R ORAI1-CFP mutants (also driven by the weak TK promoter and without co-expressed STIM1) showed frequencies of total Ca$^{2+}$ oscillations similar to those of ORAI1β-expressing cells (Fig. 5c–f), while cells expressing ORAI1 S34D phosphomimetic mutant had Ca$^{2+}$ oscillations of equal frequency to those of ORAI1-expressing cells (Fig. 5e, f).

We then designed experiments using the same system of physiological (driven by TK promoter) and exclusive (without STIM1) expression of either ORAI1-CFP or ORAI1β-CFP in HEK293 ORAI1-KO cells, to test whether different agonist-triggered oscillatory Ca$^{2+}$ signatures generated by ORAI1 and ORAI1β differentially impact the activation of two critical isoforms of NFAT with distinct Ca$^{2+}$ sensitivities, namely NFAT1 and NFAT4. With each ORAI1 isoform, we co-expressed one construct encoding either NFAT1-GFP or NFAT4-GFP fusion proteins. We used live-cell imaging to monitor NFAT-GFP isoform nuclear translocation upon stimulation with either thapsigargin or carbachol. Complete NFAT1 nuclear translocation in ORAI1-expressing cells required robust Ca$^{2+}$ entry such as that induced by thapsigargin (2 μM TG; Fig. 5g, h), while stimulation with a low concentration of agonist (10 μM Cch; Fig. 5k, l; see inset) failed to induce any detectable NFAT1 nuclear translocation across all cells studied. High concentrations of agonist (300 μM Cch) caused only modest NFAT1 translocation (Fig. 5i, j; see inset). While it was rather clear that no differences existed between ORAI1 and ORAI1β in their ability to support NFAT1 nuclear translocation (Fig. 5g–l), NFAT1 movement to the nucleus was relatively slow ($\tau$ of thapsigargin = 528 ± 52 s and 441 ± 37 s; and of carbachol = 488 ± 49 s and 529 ± 55 s for ORAI1 and ORAI1β, respectively). However, while both ORAI1 and ORAI1β supported NFAT4 nuclear translocation with similar magnitude and kinetics in response to maximal CRAC channel activation with 2 μM thapsigargin (Fig. 5m, n; $\tau$ = 432 ± 57 s and 440 ± 56 s for ORAI1 and ORAI1β, respectively), ORAI1β supported faster and more robust NFAT4 nuclear translocation compared with ORAI1 under physiological concentrations of agonist (10 μM Cch) stimulation ($\tau$ = 244 ± 18 s for ORAI1β vs 336 ± 27 s for ORAI1; Fig. 5o, p; see also Fig. 5q–s for quantification).

**Discussion**

Previous studies reported in vitro Ca$^{2+}$-dependent binding of calmodulin (CaM) to an isolated ORAI1 N-terminal peptide (residues 68–91; Fig. 1a) was curtailed when several point mutations were introduced into the ORAI1 peptide (A73E; W76E, A, or S; Y80E)[10,43] and the same mutations inhibited CDI when introduced into full-length ORAI1[10]. However, despite these findings, Ca$^{2+}$-CaM binding to an isolated ORAI1 peptide does not reflect binding to native ORAI1 channels within the physiological context of living cells, because experimental evidence using dominant-negative CaM mutants militated against CaM involvement in CDI of CRAC channels[44]. Furthermore, the crystal structure of *Drosophila* ORAI revealed the channel to be a hexamer with the key residues proposed to bind CaM (e.g., W76 and Y80) facing the pore lumen and not the cytosolic side where they could interact with CaM[45]. Several mutagenesis studies have identified critical regions in STIM1 and ORAI1 important for CDI. These include C-terminal residues 470–491 of STIM1[10,11,37,38], and the critical interplay between the three major cytosolic domains of ORAI1 (N-terminus, TM2-3 loop, and C-terminus) as well as inner pore residues[11,31,38,46,47], highlighting the complexity of the mechanisms of CDI. Some

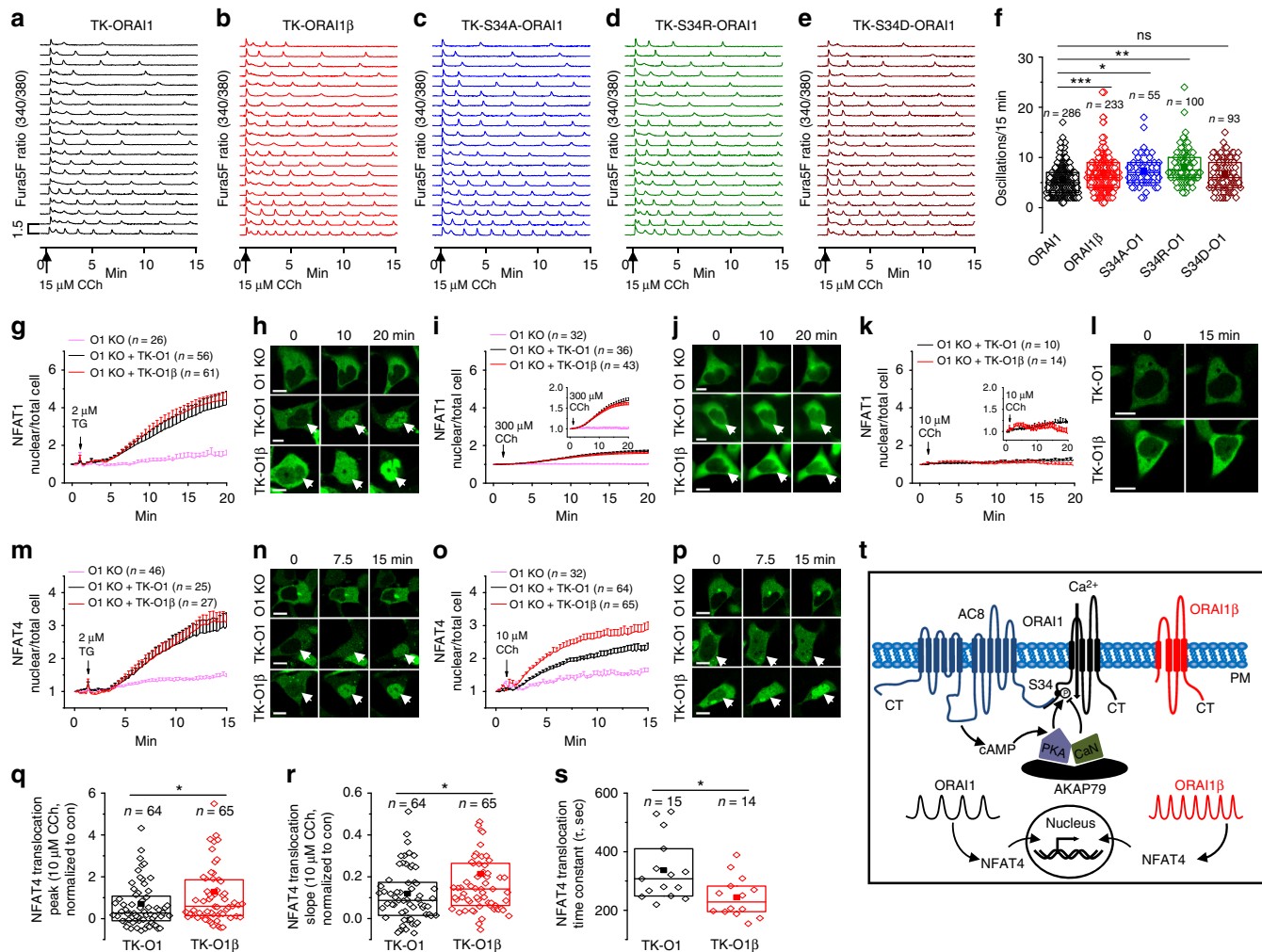

**Fig. 5** CDI regulates the frequency of ORAI1 $Ca^{2+}$ oscillations and NFAT activation. **a–f** Representative traces depicting cytosolic $Ca^{2+}$ oscillations activated by 15 μM carbachol (Cch) and measured with the dye Fura5F in ORAI1-KO cells expressing different ORAI1 cDNA plasmids driven by the weak TK promoter (without STIM1 co-expression). These are: **a** ORAI1-CFP, **b** ORAI1β-CFP, **c** S34A ORAI1-CFP, **d** S34R ORAI1-CFP, and **e** S34D ORAI1-CFP. **f** Summary data of number of $Ca^{2+}$ oscillations/15 min represented as boxplots showing the mean and median, with box representing the 25th to 75th percentile range. **g–p** Representative fluorescence traces and corresponding images of **g–l** NFAT1-GFP and **m–p** NFAT4-GFP nuclear/total ratio changes over time in response to thapsigargin (TG) and different concentrations of carbachol (Cch) in ORAI1-KO cells expressing either ORAI1-CFP (black traces), or ORAI1β-CFP (red traces) driven by the weak TK promoter. ORAI1-KO cells transfected only with NFAT-GFP (pink traces) are shown. Data represented as mean ± SEM. White arrows in picture points to NFAT-GFP nuclear translocation. Scale bar: 10 μm. Insets in (**i**) and (**k**) represent the same data on a different y scale. **q** Peak, **r** slope, and **s** time constant of NFAT4-GFP nuclear translocation in response to 10 μM carbachol in ORAI1-KO cells expressing either ORAI1-CFP, or ORAI1β-CFP. **s** Time constants could be reliably calculated only from a portion of cells. Data are represented as boxplots showing the 25th to 75th percentile range, mean and median of the peak, slope and time constant. **t** Cartoon summarizing findings from the current study. *$p < 0.05$, **$p < 0.01$, ***$p < 0.001$, two-tailed Student's $t$ test was used for (**f**, **q**, **r**, **s**)

residues in STIM1 and ORAI1 described in previous studies might represent $Ca^{2+}$ binding sites required for CDI. Furthermore, phosphorylation of serine-34 in ORAI1 might affect CDI by altering STIM1 interaction with pore-lining residues. Clearly, incorporation of previous findings with ours into a unifying model of ORAI1 CDI requires additional studies.

Our results (summarized in Fig. 5t) show that ORAI1 couples to $Ca^{2+}$-activated AC8 and cAMP signaling through a sequence (residues 26–34) in its N-terminal 63 amino acids, which are absent from its shorter translational variant, ORAI1β. We show strong associations between AC8 and ORAI1, by comparison with AC8/ORAI1β interactions. Disruption of AC8/ORAI1 interactions by mutating the AC8-binding site or the caveolin-binding site on ORAI1 and knockdown of AC8, each inhibit ORAI1 CDI without effect on ORAI1β. Maneuvers that enhance cytosolic cAMP levels including AC8 overexpression, activation

of adenylyl cyclase with forskolin or introduction of cAMP directly into the cytosol through the patch pipette, enhance ORAI1 CDI without affecting CDI of ORAI1β. We show that cAMP mediates its effects on ORAI1 CDI through PKA catalytic activity and requires PKA interaction with the anchoring protein AKAP79 at the plasma membrane. PKA directly phosphorylates ORAI1 on serine-34 to induce CDI and this effect of PKA is opposed by the signaling action of the $Ca^{2+}$-activated phosphatase calcineurin. Our findings contribute to understanding of ORAI1 inactivation by uncovering a mechanism of ORAI1 fast CDI and show that alternative translation-initiation of any channel regulates intracellular signaling.

ORAI1 and ORAI1β do not appear to heteromerize and instead form distinct homomeric channels[12]. Therefore, the implication from our findings regarding the specific signals exclusively controlled by ORAI1 are twofold: (1) forward

signaling from ORAI1 to cAMP can control specific gene transcription, most notably through activation of cAMP-responsive element binding (CREB) transcription factors, which are involved in transcription of metabolic genes[48] and (2) reverse signaling from cAMP to ORAI1 can enhance ORAI1 CDI to limit $Ca^{2+}$ entry and fine-tune ORAI1 $Ca^{2+}$ signature and NFAT activation. Our results show that enhanced CDI of ORAI1 generates low frequency $Ca^{2+}$ oscillations driven by low physiological concentrations of a G protein-coupled receptor (GPCR) agonist. While $Ca^{2+}$ oscillations result from $Ca^{2+}$ release through $IP_3$ receptors embedded in the ER membrane, $Ca^{2+}$ entry through CRAC channels is required to replenish the ER reserves and sustain these oscillations[41,49]. Therefore, the low frequency of ORAI1-mediated $Ca^{2+}$ oscillations (compared with ORAI1β) is the direct result of CDI.

The major transcription factors activated by $Ca^{2+}$ originating from CRAC channels are NFAT isoforms (NFAT1-4)[39,50]. NFAT1 and NFAT4 isoforms possess distinct activation kinetics; whereas NFAT4 shows fast nuclear localization, NFAT1 nuclear translocation is slow[15–17]. Here we show that NFAT1 requires a robust CRAC-mediated $Ca^{2+}$ signal for nuclear translocation. Even when supramaximal concentrations of GPCR agonist are used, only slight NFAT1 movement to the nucleus is detected and this translocation is robust only when maximal activation of CRAC channels is achieved through pharmacological depletion of ER stores with thapsigargin. However, NFAT4 migration to the nucleus is fast and is readily observed when low physiological concentrations of GPCR agonist are used to stimulate cells. Previous studies showed that different frequencies of oscillatory $Ca^{2+}$ signals activate different families of $Ca^{2+}$-sensitive transcription factors[51]. Here we show that an isoform of the same family of transcription factors is differentially activated depending on the extent of ORAI1 CDI. NFAT1 nuclear translocation, which requires robust $Ca^{2+}$ signals is supported equally by both ORAI1 and ORAI1β. However, NFAT4, which is activated by low physiological concentrations of GPCR agonizts, is preferentially activated by ORAI1β. Previous studies showed that unlike NFAT2 and NFAT4 isoforms, which regulate physiological functions such as cell proliferation and secretion, NFAT1 is pro-apototic and its activation causes cell cycle arrest and death[16,52]. Therefore, the implication from our data is that CDI is not merely a means to curtail pathophysiological $Ca^{2+}$ overload-mediated cellular toxicity under conditions of supramaximal stimulation, but rather CDI acts as a modulator of signaling and transcription factor activation that enhances the diversity of $Ca^{2+}$ signaling pathways under physiological conditions. Thus we show that CDI differentially regulate transcription factor activation.

AC8 expression is elevated in cells of neuro-epithelial lineage such as HEK293 cells. Whether ORAI1 CDI is more prominent in these primary cell types to mediate specific physiological functions requires future clarifications. Based on our work, it is reasonable to propose that the relative proportions of ORAI1 versus ORAI1β and the abundance of AC8 in a given cell type alters cellular signaling to fine-tune signaling and transcription. The distinction between ORAI1 and ORAI1β and their relative proportions compared with each other and to AC8 in primary cell types and their impact on cell signaling, transcription and physiological function awaits investigation.

## Methods

### Generating ORAI1 knockout HEK293 cells using CRISPR/Cas9. ORAI1 sequence-specific guide RNAs were inserted into the lentiCRISPR v2 vector (Addgene, Plasmid #52961) with the BsmBI restriction site to create a gRNA-Cas9-encoding plasmid. The complete list of primers used is included in Supplementary Table 2. HEK293 cells (ATCC, CRL-1537) were transfected with the gRNA-Cas9 plasmid using the Amaxa Nucleofector (Lonza, Nucleofector™ 2b Device) and the cell line Nucleofector Kit (Lonza, VCA-1003) according to the manufacturer's

protocol (Q-001). Forty-eight hours after transfection, cells were cultured in DMEM supplemented with 10% fetal bovine serum, penicillin, and streptomycin containing puromycin (2 μg/ml) (Gemini Bio Products, West Sacramento, CA) in 5% $CO_2$ at 37 °C. Six days after puromycin selection, cells were collected and seeded at one cell per well into 96-well plates. Disruption of the ORAI1 gene in individual colonies was detected using the Guide-it Mutation Detection Kit (Clontech Laboratories, 631443) and confirmed by sequencing, as well as western blotting and functional $Ca^{2+}$ imaging assays. The ORAI1 knockout HEK293 cell line generated was named ORAI1-KO HEK293. The ORAI1 guide RNA used in this study was: 5′-GTTGCTCACCGCCTCGATGT-3′.

### Generating STIM1/2 knockout HEK293 cells. The procedures of generating STIM1 knockout, STIM2 knockout, and STIM1/2 double knockout HEK293 cell lines are similar to those for ORAI1-KO HEK293 cells, except for guide RNAs. Guide RNA for STIM1: 5′-TGATGAGCTTATCCTCACCA-3′. Guide RNA for STIM2: 5′-AGATGGTGGAATTGAAGTAG-3′.

### Constructs and transfections. To generate a vector exclusively expressing long ORAI1, we used a wild-type ORAI1-CFP cDNA clone in which we exchanged the relatively weak native Orai1 Kozak sequence upstream of the first ATG start site (TGCTCCATG) with a stronger Kozak sequence (GCCACCATG), thus forcing the translational machinery to exclusively start at the first ATG. For a vector exclusively expressing ORAI1β, the first ATG start site of ORAI1-CFP cDNA was mutated to GCG (alanine), leaving only the start site at methionine-64[12,13] (see Supplementary Table 2 for primers used). To obtain non-tagged AC8 from YFP-AC8, we used Clontech in-fusion HD cloning method and designed two primers (see Supplementary Table 2) to get the YFP tag out, gel purified the PCR product and fuse it to get non-tagged AC8. The same cloning method was used to generate Δ1-107-AKAP79-GFP, lacking the first 107 residues (primers listed in Supplementary Table 2). To swap different fluorescent tags between constructs, we used a first set of primers (pair 1; Supplementary Table 2) to amplify the fluorescent protein we need to have in the final construct and used the second primer pair (pair 2; Supplementary Table 2) to remove the original tag and amplify the vector, then gel purified both PCR products and used Clontech in-fusion HD kit to fuse both products together. A similar strategy was used to swap CMV promoter with TK promoter (pair 1 and pair 2; Supplementary Table 2), except that both PCR products were cut with Nhe1 before ligation. HEK293 cells and ORAI1 knockout (ORAI1-KO) and STIM1/STIM2 double knockout HEK293 cells were grown in Dulbecco's modified Eagle's medium (DMEM) supplemented with 10% (v/v) heat-inactivated fetal bovine serum, penicillin (100 U/ml), and streptomycin (100 mg/ml) and maintained in a humidified 95% air and 5% $CO_2$ incubator at 37 °C. In total, $1 \times 10^6$ cells were transfected with Lipofectamine 2000 (Invitrogen) or Nucleofector II Device (Amaxa Biosystems) and placed onto 30-mm round glass coverslips (#1.5 thickness) in 6-well plates. For patch clamp recordings, 4.0 μg of a plasmid encoding eYFP-STIM1 and 1.0 μg of a plasmid encoding either ORAI1-CFP or ORAI1β-CFP were used. For siRNA against AC8 experiments, siRNA oligonucleotides, and non-targeting control siRNA were transfected at a final concentration of 300 nM. For siRNAs against PPP3R1 and PPP3R2 experiments, siRNAs oligonucleotides, and non-targeting control siRNA were transfected at a final concentration of 250 nM. Cells were used 24–48 h post transfection. The siRNA used throughout the study are reported in Supplementary Table 3.

### Patch clamp electrophysiology. We used ORAI1-KO cells transfected with 4.0 μg of a plasmid encoding eYFP-STIM1 and 1.0 μg of a plasmid encoding either ORAI1-CFP or ORAI1β-CFP. This 4:1 STIM1/ORAI1 transfection ratio was used throughout the study to ensure that STIM1 proteins are not limiting and maximal CDI is attained[10,36]. To reliably detect large CRAC currents, cDNA plasmids encoding STIM1, ORAI1, and ORAI1β were driven by the robust cytomegalovirus (CMV) promoter and CRAC channel recordings were performed 24–48 h post transfection. HEK293 cells were seeded onto glass coverslips in the 6-well plates 24 h before electrophysiological recordings. We used fluorescence microscope to identify transfected cells with sufficient intensity of fluorescence of tagged protein. Traditional patch clamp electrophysiological recordings were carried out using an Axopatch 200B and Digidata 1440A (Molecular Devices) as previously published[53,54]. Pipettes were pulled from borosilicate glass capillaries (World Precision Instruments) with a P-1000 Flaming/Brown micropipette puller (Sutter Instrument Company) and polished using DMF1000 (World Precision Instruments). Resistances of filled pipettes were 1.5–4 MΩ. Under whole-cell configuration, only cells with <8 MΩ series resistances and tight seals (> 8 GΩ) were chosen to perform recordings. Clampfit 10.3 software was used for data analysis. Bath solution: 115 mM Na-methanesulfonate, 10 mM CsCl, 1.2 mM $MgSO_4$, 10 mM Hepes, 20 mM $CaCl_2$, and 10 mM glucose (pH 7.4 with NaOH). Pipette solution: 135 mM Cs-methanesulfonate, 10 mM EGTA, 8 mM $MgCl_2$, and 10 mM Hepes (pH 7.2 with CsOH). Immediately after break-in and before CRAC currents have developed, we ran a stimulus protocol consisting of families of 150 ms voltage steps (from + 30 mV holding potential to −120, −100, −80, and −60 mV). We allowed 2 s interval between steps. At the beginning of each pulse, a 2.5 ms voltage step to 0 mV was used to eliminate residual capacitive artifacts from cell capacitance (Supplementary Fig. 1f). This first voltage step recording was used for leak

subtraction. Next, a 250 ms voltage ramp from $+100$ to $-140$ mV was administered every 2 s until CRAC currents reached steady-state level (usually within 150–300 s). Once steady state for CRAC current activation was achieved, CDI was determined by performing a second round of voltage steps, using the same protocol used during the first round described above. CDI was quantified as the remaining current measured at 146 ms from the peak current at the beginning of the pulse. The time course of ORAI1 CDI was fitted with a bi-exponential function: ($I = I_0 + A_1 e^{-t/\tau 1} + A_2 e^{-t/\tau 2}$); $I$ is current, $I_0$ is steady-state current, $A_1$ and $A_2$ are amplitudes, and $\tau 1$ (fast) and $\tau 2$ (slow) are time constants of inactivation. Supplementary Table 4 shows statistical analysis of all patch clamp data reported in each figure panel.

**Measurements of Ca²⁺ oscillations.** For Ca²⁺ oscillations, we used the lower affinity indicator Fura-5F [$K_d = 400$ nM] to better resolve oscillations and to ensure that the data are well below dye saturation; most of the ratios were < 2.0, well below the average observed Rmax of 5.6. Glass coverslips with attached HEK293 cells were mounted in a Teflon chamber and incubated in DMEM with 4 μM Fura-5F-AM at 37 °C incubator for 40 min. After incubation, cells were immediately rinsed three times with a Hepes-buffered salt solution (140 mM NaCl, 4.7 mM KCl, 1.13 mM MgCl₂, 10 mM HEPES, 2.0 mM CaCl₂, and 10 mM glucose, with pH 7.4 adjusted by NaOH) and incubated at room temperature for 10 min prior to imaging. Then glass coverslips with attached HEK293 cells were mounted onto a Nikon TS100 inverted microscope equipped with a ×20 Fluor objective (0.75 numerical aperture), and fluorescence images were recorded and analyzed with a digital fluorescence imaging system (InCyt Im2, Intracellular Imaging Inc.) equipped with a light-sensitive charge-coupled device camera (Cooke PixelFly, Applied Scientific Instrumentation). Fura-5F fluorescence was monitored by alternately exciting the dye at 340 and 380 nm and collecting the emission at 510 nm. Changes in intracellular calcium are represented as the ratio of Fura-5F fluorescence due to excitation at 340 nm to that due to excitation at 380 nm (ratio, F340/F380). The ratio changes in fields of Fura-5F-loaded cells were collected from multiple regions of interest (ROIs), with each ROI representing an individual cell. Typically, 10–30 ROIs were monitored per experiment.

**Ca²⁺ measurements.** Ca²⁺ measurements were performed as above with few minor changes. Coverslips with attached cells were mounted in a Teflon chamber and incubated at 37 °C for 1 h in culture medium (DMEM with 10% FBS) containing 2 μM Fura2-AM (Molecular Probes, Eugene, OR). Cells were then washed and bathed in Hepes-buffered saline solution (in mM): 140 NaCl, 1.13 MgCl₂, 4.7 KCl, 2 CaCl₂, 10 D-glucose, and 10 HEPES, with pH adjusted to 7.4 with NaOH for 10 min before Ca²⁺ was measured. Fluorescence images of several cells were recorded and analyzed with a digital fluorescence imaging system (InCyt Im2, Intracellular Imaging, Cincinnati, OH). Fura2 fluorescence at an emission wavelength of 510 nm was induced by excitation of Fura 2 alternately at 340 and 380 nm. The ratio of fluorescence at 340 nm to that at 380 nm was obtained on a pixel-by-pixel basis. All experiments were performed at room temperature.

**FRET measurements.** FRET signals were measured between transiently expressed N-terminally tagged YFP-ORAI1/β and CFP-AC8/AKAP, or between C-terminally tagged ORAI1/β-CFP and STIM1-YFP using a Leica DMI 6000B inverted automated fluorescence microscope equipped with CFP (438Ex/483Em), YFP (500Ex/542Em), and FRET (438Ex/542Em) filter cubes. For ORAI1/β and STIM1 constructs, which show high levels of expression and relatively fast interactions on store depletion, FRET images were captured every 20 s for a total of 10 min. For experiments measuring FRET interactions between ORAI1/β and AC8/AKAP, AC8, and AKAP constructs show low expression levels and only constitutive interactions with ORAI1/β. For assessing potential long-term interactions between ORAI1/β and AC8/AKAP induced by agonist stimulation, experiments were performed for a total of 40 min. As such, images were captured every 10 min to minimize potential photobleaching.

Three sets of images (CFP, YFP, and FRET) were collected at room temperature using a ×40 oil objective (N.A.1.35; Leica) at each time point and processed using SlideBook 6.0 software (Intelligent Imaging Innovations). Exposure times for the CFP, YFP, and FRET channels were 1000 ms. Three-channel corrected FRET was calculated using the formula: FC = IDA − Fd/Dd × IDD − Fa/Da × IAA. IDD, IAA, and IDA represent the intensity of the background-subtracted CFP, YFP, and FRET images, respectively. FC is equal to the corrected energy transfer. Fd/Dd is the experimentally measured bleed-through of CFP through the FRET filter (0.50), and Fa/Da is the experimentally measured bleed-through of YFP through the FRET filter (0.025). The E-FRET method was used to analyze 3-cube FRET images as described by ref. [55] using the formula: Eapp = Fc/(FC + G × IDD), where $G$ is the instrument specific constant, which was determined by measuring the CFP fluorescence increase after YFP acceptor photo-bleaching using HEK293 cells transiently transfected with eYFP-eCFP calibrator construct. The value of $G$ was calculated and is 1.9 ± 0.1. For all E-FRET summary data, the regions of interest were close to the plasma membrane, and only cells with similar YFP/CFP ratios were used for E-FRET analysis. For quantification of co-localization between AC8 (or AC8M1 mutant) and ORAI1, ROIs were drawn on the perimeter of cells and

the Pearson's correlation coefficient was calculated using Image J software and the Coloc 2 plugin.

**Pull-down and phosphorylation assays.** Cells were washed with chilled PBS and lysed in 400 μl RIPA buffer (Sigma-R0278) containing cocktail of protease inhibitors (Invitrogen-A32975) and phosphatase inhibitors (Thermo-88667). To probe for PKA-mediated ORAI1 phosphorylation, cells were incubated with or without 10 μM Forskolin (Sigma-F6886) for 15 min at room temperature before being scraped and lysed in chilled RIPA buffer. Co-immunoprecipitation was performed using Dynabeads co-immunoprecipitation kit (Invitrogen-14321D). According to the manufacturer protocol, the beads were coated with mouse anti-GFP antibody (R and D Systems-MAB42401) to immunoprecipitate eGFP-ORAI1. Equal amount of protein was added to the antibody-coated beads and manufacturer protocol was followed to elute the protein complex. Each step was performed at 4 °C. The eluted protein was resolved in 4–12% gel (NuPAGE Bis-Tris precast gels, Life Technologies-NP0321) and analyzed using western blotting with anti-ORAI1 (Sigma-O8264, 1:4000) or anti-phosphoserine (Millipore-AB1603, 1:1000). A similar protocol was used for pull-down assays between untagged AC8 and either ORAI1-CFP or ORAIβ-CFP. Beads were coated with mouse anti-GFP antibody to immunoprecipitate eGFP-ORAI1, followed by elution of the protein complex and western blotting with anti-ORAI1 or anti-AC8 antibodies.

**Western blots.** Cells were washed with chilled PBS and lysed in 100 μl RIPA buffer (Sigma-R0278) containing cocktail of protease inhibitors (Invitrogen-A32975). 20 μg of protein extract was loaded on a 4–12% gel (NuPAGE Bis-Tris precast gels, Life Technologies-NP0321) and transferred to a polyvinylidene difluoride membrane. The membrane was incubated in 0.1% TBST with 5% non-fat dry milk (Bio-Rad-1706404) for 2 h at room temperature. For probing with anti-phosphoserine antibody, the membrane was incubated for 2 h at room temperature in 0.05% TBST with 5% BSA (VWR-97061416). Then the blot was incubated overnight at 4 °C with primary anti-phosphoserine antibody. Primary antibodies were used at the following dilutions: rabbit-anti-ORAI1 (Sigma-O8264) 1:4000; rabbit-anti-Phosphoserine (Millipore-AB1603) 1:1000, rabbit-anti-ADCY8 (AC8; Proteintech-55065-1-AP) 1:4000, and mouse anti-GFP (R and D Systems-MAB42401). The membrane was washed with 0.1% TBST for 10 min (three times) at room temperature and followed by incubation in secondary antibody for 2 h at room temperature. Secondary antibodies: HRP-conjugated anti-Mouse IgG (Invitrogen A27025), dilution 1:4000; and HRP-conjugated anti-rabbit (Jackson Immuno Research-323005021), dilution 1:5000. The membrane was washed with 0.1% TBST for 10 min (three times) at room temperature. Then the peroxidase activity was detected by ECL prime western blotting detection regent (GE healthcare-RPN2232). Densitometry analysis of the bands on membranes was performed using ImageJ.

**NFAT nuclear translocation assays.** For NFAT-GFP nuclear translocation experiments, $1.5 \times 10^6$ cells were transfected with 1.0 μg of NFAT-GFP plasmids and 0.5 μg of TK-ORAI1/β plasmids using Lipofectamine 2000. Fluorescence imaging was performed on a Leica DiM8 confocal microscope with a ×40 objective. After baseline recordings, carbachol (Cch) or thapsigargin (TG) were added at 1 min. A ratio of nuclear NFAT-GFP to total NFAT-GFP was calculated on a cell by cell basis.

**Quantitative PCR.** RT-qPCR was performed following established methods. Cells were first homogenized in TRIzol (Invitrogen) and total RNA isolated according to the manufacturer protocol. Total RNA was then quantified using a nanodrop 2000 spectrophotometer (Thermo Scientific). A volume corresponding to 1 μg of total RNA was reverse transcribed into cDNA using the High-Capacity cDNA Reverse Transcription Kit (Applied Biosystems). Total cDNA (6 μl) was then added to SYBR Green qPCR Master Mix (Applied Biosystems), and the corresponding primers added resulting in a 20 μl reaction. Samples were then loaded into a 96-well plate and data from the resulting reaction was collected and quantified using the QuantStudio 3 real-time PCR system (Applied Biosystems). The PCR protocol for all reactions began with an initial activation of 2 min at 50 °C step followed by a 95 °C 2-min melt step. The initial steps were followed by 40 cycles that began at 95 °C for 15 s followed by 54.3 °C for 15 s and 72 °C for 30 s. After completion of 40 cycles, a standard melt curve was generated to ensure primer specificity. All primer sequences used are provided within supplementary table 2. Analysis of target and control samples was carried out using the comparative $C_t$ method. Experimental samples were normalized to the reference gene glyceraldehyde-3-phosphate dehydrogenase (GAPDH). All data were generated in technical and biological triplicates to ensure reproducibility.

**Quantification.** ImageJ software was used for quantifying protein levels from western blots. InCytim2 V5.35 software was used for quantifying fluorescence signals from Ca²⁺ imaging assays. SlideBook 6.0 software was used for quantifying FRET fluorescence signals. LAS X Life Science Microscope Software was used for quantifying NFAT1-GFP and NFAT4-GFP nuclear translocation.

**Statistical analysis**. All data are expressed as mean ± SEM. Statistical analyses comparing two experimental groups were performed using two-sample $t$-test with Origin 9.0 software. Statistical analyses comparing more than two experimental groups were performed using one-way ANOVA with Origin 9.0 software. See Supplementary Table 4 for quantification of all patch clamp electrophysiology data. Throughout the figures *, **, and *** indicates $p$-values of $< 0.05$, $< 0.01$, and $< 0.001$, respectively. Differences were considered significant when $p < 0.05$.

**Reporting summary**. Further information on experimental design is available in the Nature Research Reporting Summary linked to this article.

## Data availability

Data supporting the findings of this manuscript are available from the corresponding authors upon reasonable request. A reporting summary for this Article is available as a Supplementary Information file. The source data underlying Figs. 1d, 1g, 1j, 1n, 1o, 2c, 2e, 2i, 2l-2n, 3c, 3f, 3i, 3l, 3p, 4c, 4i, 4k, 4o, 4s, 5f, 5q-s and Supplementary Figs. 1e, 1g, 1h, 1k, 1n, 2c, 2f, 2i, 2l, 3c, 3f, 4c, 4f, 4i, 5d, 5e, 5f, 6g, 6k, 7d, 7g, 8c, 9d, 9h, 10a, 10b, 10c, 10d, 11a, 11b, 11d and uncropped gels and blots are provided as a Source Data file.

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

## Acknowledgements

The authors acknowledge Dr. Tamas Balla (NIH) for the gift of the Thymidine kinase (TK) promoter. Dr. Trevor J. Shuttleworth (University of Rochester) for the STIM1-pVenus and T389A and T389E STIM1-pVenus mutants. Dr. Dermot M. F. Cooper (Cambridge University) for tagged AC8 and AC8M1. Our study was supported by the National Heart, Lung, and Blood Institute (R01-HL123364 and R01-HL097111 to M.T.), National Institute on Aging (R21-AG050072 to M.T.), by National Institute of General Medical Sciences (R01-GM109279 and R01-GM120783 to D.L.G.), and by the Intramural Research Program of the National Institutes of Health, National Institute of Environmental Health Sciences (ES-090087) to J.W.P.

## Author contributions

X.Z. contributed to the design of experiments, performed patch clamp electrophysiology and calcium imaging experiments, analyzed the data, generated figures, and contributed to the writing of the manuscript. T.P. performed pull-down assays, westerns and phosphorylation assays. R.Y. and S.E performed real-time PCR experiments, FRET and NFAT nuclear translocation assays. P.X. engineered most plasmids and generated all CRISPR/Cas9 knockout cell lines. R.N. performed the FRET experiments. M.J. performed western blots experiments. S.W. engineered the original ORAI1 and ORAI1β clones driven by the TK promoter. C.D. and M.G. contributed to discussions. N.H., D.L.G., and J.W.P. contributed to experimental design, and editing of manuscript. M.T. conceived the hypothesis, designed the experiments, supervised the research, interpreted the data and wrote the manuscript.

## Additional information

**Competing interests:** The authors declare no competing interests.

