## [Peer Review File · Nature Communications]

Reviewers' Comments:

Reviewer #1:

Remarks to the Author:

In this manuscript, the authors have pursued the mechanism of Ca²⁺ dependent inactivation of Orai1-mediated Ca²⁺ currents, which are activated by STIM1 following depletion of ER Ca²⁺ stores. A number of previous studies have been carried out over the last 10 years to try to understand how this process occurs, identifying various elements of the Orai channel and in its activator, STIM1, that are involved in CDI. How these various structural features support CDI has not been very clear, however.

Here, the authors have studied CDI by exploiting the fact that an alternative-initiation site variant of the Orai1 channel, Orai1-beta, largely lacks CDI. They have used electrophysiology and pharmacology, pull-down assays and FRET measurements to obtain results indicating that the Ca²⁺-activated adenylyl cyclase AC8 is associated with the N-terminal region (lacking in Orai1-beta) of Orai1; that AC8 is activated by Ca²⁺ influx through Orai1 to produce cAMP and thereby activate PKA localized to the complex by the A-kinase anchoring protein AKAP79; and that phosphorylation of a serine residue in the N terminal region of Orai1 mediates this channel's inactivation.

In addition, the authors provide results suggesting a possible role for regulated Orai1 CDI in cellular function: the naturally-occurring Orai1-beta variant produces more frequent oscillations in cytosolic Ca²⁺ in response to application of IP3R-pathway activating agonist (carbachol), and when the Orai1 variant is mutated at the N-terminal domain serine supporting CDI so that Orai1 CDI is prevented/reduced, the mutated Orai1 supports more frequent Ca²⁺ agonist-activated Ca²⁺ oscillations, similar to Orai1-beta. Significantly, the more frequent oscillations for Orai1-beta correlate with a greater initial rate and total amount of cytosol-to-nucleus translocation of the Orai/calmodulin/calcineurin-regulated transcription factor NFAT4. In contrast, no differential nuclear translocation was observed for NFAT1 translocation.

The experimental results supporting these conclusions are strong and are actually quite a bit more comprehensive than indicated above. The paper is clearly written. The findings are certainly important because they open up new avenues for investigation, both in CDI and in understanding differential control of signaling to the genome.

Specific points:

A significant problem for the way the story is presented in the manuscript is that Orai1-beta does clearly exhibit significant CDI, albeit much less than does Orai1. Orai1-beta CDI is eliminated, as for Orai1, when extracellular Ca²⁺ is replaced by Ba²⁺. Thus there is more going on, in regard to CDI, than is presented in the manuscript. This should be clearly addressed in the writing.

In the wording of the manuscript, I would suggest toning down the idea that this is the first major step forward in understanding CDI of Orai channels (even though there is a case to be made that this is true). A great deal of work has been done previously, without a lot of forward advance. But based on the present manuscript, those findings will ultimately need to be incorporated into a more complete description of CDI, particularly downstream from the phosphorylation step identified here. It may also be that CDI of Orai channels is more complicated than presented here, and it could also be different in different cellular contexts (excitable versus non-excitable cells studied here).

Activation of Orai1 by the STIM1 activation domain (SOAR, CAD) peptide is known to produce Orai1 currents that do not exhibit fast CDI: how could that intriguing finding be incorporated into the mechanism reported here?

Reviewer #2:

Remarks to the Author:

The manuscript confirms the previously reported interaction between adenylyl cyclase 8 (AC8) and the ORAI1 N terminus. It further establishes that there is a pathway in HEK293 cells leading from AC8, through cyclic AMP and protein kinase A (PKA) positioned by AKAP79, to phosphorylation of ORAI1 S34. The phosphorylation of S34 enhances fast Ca^{2+} -dependent inactivation (CDI) of ORAI1 currents. Fast CDI correlates with fewer cytosolic Ca^{2+} transients in HEK293 cells stimulated with low concentrations of the agonist carbachol, and there is a corresponding difference in NFAT4 nuclear translocation between cells expressing ORAI1, which is subject to the AC8 regulatory pathway, and cells expressing ORAI1 β , which is not.

The authors tackle a difficult question, and make a compelling case that an AC8 > cAMP > PKA > S34 phosphorylation pathway is involved in regulating fast CDI in HEK293 cells. The experiments are technically flawless, and the manuscript is very clearly written. An additional important finding is that fast CDI, usually measured under extreme conditions of Ca^{2+} influx, correlates with significant changes in cellular Ca^{2+} signaling under physiological conditions. The latter point deserves further exploration, though as part of a separate paper. Some questions of interpretation remain.

(1) Does the AC8 pathway account completely for fast CDI in HEK293 cells? Comparison of the currents recorded with BAPTA or with EGTA in the intracellular solution has been the defining measure of fast CDI triggered by Ca^{2+} binding to an intracellular site near the channel. Here, the degree of protection of ORAI1 current with BAPTA (74% remaining current for the -120 mV step, compared to 49% with EGTA) is much larger than for ORAI1 R31A-R33A, siAC8, H89, Ht-31 peptide, or ORAI1 S34A (where remaining current ranges from 53%-56%). The divergence suggests that another process might account for a large fraction of fast CDI in HEK293 cells.

(2) Are the AC8 pathway and phosphorylation of ORAI1 S34 required for fast CDI? If the question is taken in a literal sense, the answer is no. A considerable degree of fast inactivation is seen with ORAI1 S34A when Ca^{2+} is the current carrier (Figure 4e and 4i), and is absent when Ba^{2+} is the current carrier (Supplementary Figure 7b), satisfying a criterion for Ca^{2+} dependence. However, it might be that Ca^{2+} acts in this case at a site not accessible to BAPTA (Scrimgeour et al. 2012 *Biochem. J.* 441,743; Scrimgeour et al. 2014 *Biochim. Biophys. Acta* 1838, 1281), and, if so, the literal interpretation might include mechanisms outside the conventional notion of fast CDI. Rephrasing the question to ask whether a second intracellular site near the channel is linked to fast CDI, an experimental test would be whether BAPTA still has a protective effect in the case of ORAI1 S34A.

(3) The manuscript does not offer evidence that it is Ca^{2+} entering through the ORAI1 channel that, within a few milliseconds, activates the AC8 pathway controlling fast CDI. A plausible alternative is that a pre-existing level of S34 phosphorylation sets ORAI1 susceptibility to a distinct fast CDI process. In fact there is background S34 phosphorylation in the absence of forskolin in Figure 4k. (It is difficult to assign functional relevance or irrelevance to the apparent level of background phosphorylation, since it has not been determined how many of the six S34 sites in a channel need to be phosphorylated to enhance fast CDI.) Forskolin increases S34 phosphorylation, as intended, and in fact all the tested conditions designed to manipulate the AC8 pathway could potentially change the pre-existing level of S34 phosphorylation.

(4) There is a further reason to be concerned about the direct connection between Ca^{2+} entering through the channel and the AC8 pathway. The turnover number reported for engineered soluble mammalian AC (30 s^{-1} , Dessauer and Gilman 1997 *J. Biol. Chem.* 272, 27787) seems too sluggish to drive a process whose τ_{fast} is 5 ms. Ideally the issue would be settled by detecting a rapid increase in S34 phosphorylation, under the relevant conditions for fast CDI, correlated with the amount of Ca^{2+} influx, and on the millisecond time scale. The ideal experiment would be prohibitively difficult at present, but the authors should at least note that they are proposing a specialized mechanism that might entail, for example, AC8 poised for a single enzymatic turnover and direct transfer of cyclic AMP to PKA tethered nearby.

(5) Statements in the title, abstract, and text that fast CDI has a role in shaping transcription are

unsupported, since no data on transcription are presented in the manuscript.

(6) Discuss whether it is likely there are cell-type differences in fast CDI. HEK293 cells are of neuroepithelial origin, and AC8 expression is high in neuroepithelial-lineage cells. Is the same mechanism prominent in other cell types?

(7) The average numbers of Ca²⁺ oscillations in the example traces of Figure 5a, 5c, and 5e deviate substantially from the means plotted in Figure 5f, in a way that makes ORAI1 and ORAI1 S34D appear less active, and ORAI1 S34A more active. One way to obtain a more representative set of example traces would be to present four traces selected randomly from the lowest quintile for each condition, four traces from the next quintile, and so on.

Reviewer #3:

Remarks to the Author:

Overall, the findings of Zhang et al, regarding the mechanism of Orai1 CDI are potentially interesting to the field. Their main finding is that Orai1 CDI is controlled, at least in part, by phosphorylation of the Orai1 N-terminal tail (likely at S34) by PKA. They propose that PKA is activated by cAMP generated from Ca²⁺-dependent adenylyl cyclase AC8 and recruited to the membrane by AKAP. Although interesting, the main problem with the study is that these effects are very small and the authors overstate their conclusions regarding the centrality of the PKA effect for inactivation. The AC8/PKA model is framed as THE sole mechanism for CDI but the results here do not show that. The conclusions of this study will have to be significantly scaled down to be appropriate to the results shown.

1) The majority of the changes shown in CDI seen are in the 10-20% range (the key experiments in Figures 2-4 make this very clear). The Y-axis scales used for the plots and the high number of cells in each group (10-25 cells) make the effects appear highly statistically significant. However, the authors overstate the contribution of this pathway for CDI when there are clearly many other contributing factors.

2) Orai1b still inactivates substantially even without the proposed phosphorylation site, none of the manipulations shown are able to recapitulate the entire difference in CDI between Orai1 and Orai1b, and the connection between their AC8/PKA/AKAP effects and S34 mutants appears to be tenuous. The authors need present this pathway as one contributor to CDI instead of THE mechanism.

3) Could the changes in CDI arise from changes in STIM1-binding? STIM1 binding is essential for CDI (see relevant papers by Lewis, Rychkov) so any changes in the binding affinity of Orai1b channels lacking the N-terminus could influence the level of CDI. There is no data at all shown on this critical point. It is essential for this study to show that STIM1 binding is not altered in Orai1b as well as by manipulations (e.g. AC8/PKA pharmacology, S34 mutants).

4) If Orai1 CDI is modulated by phosphorylation of S34, why do mutations of S34 to D/E residues have no effect on CDI? Figure 4j shows that WT Orai1 is not phosphorylated at baseline without addition of forskolin, which suggests that a phosphomimetic mutation should enhance CDI. It would seem from this result that the effects at S34 on CDI are independent of the AC8/PKA pathway.

5) CDI is acutely sensitive to the local Ca²⁺ concentration around individual CRAC channels. Mullins et al, (JGP, 2016) have shown that some N-terminal Orai1 mutations elicit large changes in the single channel conductance of Orai1 channels. If this is also the case for the mutants and the Orai1b isoform of this study, then amount of local calcium at the mouth of the channel would be very different in Orai1b vs Orai1 channels (or the phosphorylation mutants). The authors would need to rule out this key possibility using noise analysis or other methods.

6) If CDI is defective in Orai1b compared to Orai1 as concluded, why are the kinetics of inactivation faster as measured by tau 1 and tau 2? This does not make sense.

7) In supplemental figure 8, a statistically significant difference in the "percentage of traces with one spike event" seems to be arbitrary, would be better to show histogram of the number of spike events in Orai1 vs. Orai1b cells. Also, how are "plateau events" defined? The authors will need to use more quantitative measures such as average duration or tau of decay of the events.

8) The discussion only alludes to the CaM model of CDI - the authors should also mention the proposed roles of the N-terminus (Frischauf 2011), TM2-3 loop (Srikanth 2010, Frischauf 2011), C terminus (Lee 2009), inner pore residues (Mullins 2016)...etc. Can their model explain the differences in CDI between Orai1/2/3 isoforms? How could the modulation of the N-terminus mediate CDI? It is clear that other regions also play roles in CDI, so their AC8/PKA model should be not be framed as the sole mechanism.

Minor typos:

- Second sentence of introduction is missing the word "current" at the end.
- Page 8, second paragraph line 8: should be "live-cell imaging".

Reviewer #4:

Remarks to the Author:

The main finding is important, the study is detailed and control experiments are reasonable.

One somewhat surprising finding is that the phenomenon involving Ca²⁺ entry, Ca²⁺-dependent cAMP production and PKA-dependent phosphorylation of ORAI1 is sufficiently fast to produce CDI (on time scale of few milliseconds). This could be perhaps discussed in conjunction with the known kinetics of cAMP production/PKA phosphorylation. In future (not necessarily for this paper) cAMP uncaging could be used to further characterise the surprisingly fast Ca²⁺-cAMP-PKA response.

It would be useful to test the sensitivity of NFAT4 nuclear translocation to cAMP-increasing agents (in cells expressing TK-ORA1, TK-ORAI1 β and one of S34 mutants)

Minor point: P.6. Please specify "Online software tools..."

We would like to sincerely thank all four reviewers for the careful and critical review of our manuscript. Although reviewers found our study to be carefully performed, strong and novel, they have identified an issue with the interpretation of data and have requested few additional controls to strengthen our conclusions.

All reviewers were unanimous on two points in the manuscript: 1) that the role of AC8 in ORAI1 inactivation we have identified should be framed as a novel mechanism of ORAI1 fast inactivation, and not THE sole mechanism of inactivation. 2) That the signaling events triggered by AC8 leading to ORAI1 phosphorylation on serine-34 are unlikely fast enough to be the actual trigger of CDI and that one or more Ca²⁺-binding sites are involved. We have humbly heeded the reviewers' assessment and concur that the mechanisms of ORAI1 CDI are complex with lots going on. Accordingly, we have made changes in the abstract and text to reflect this fact and have made it very clear early in the results section that the involvement of AC8 alone CANNOT account for all of the extent of ORAI1 CDI. We also show data (from an experiment suggested by reviewer 2) indicating that additional Ca²⁺ binding site(s) likely involved in ORAI1 CDI. Furthermore, we have performed additional experiments suggested by reviewer 3, to further support of our conclusions. These data are included in three additional supplementary figures (suppl. Fig. 8-10). Revisions and changes to the manuscript are clearly highlighted and below we respond point-by-point to reviewers' comments. To facilitate reading of this document, below we highlighted the reviewers' original comments in blue followed by our response (in black).

Reviewer #1 (Remarks to the Author):

In this manuscript, the authors have pursued the mechanism of Ca²⁺ dependent inactivation of Orai1-mediated Ca²⁺ currents, which are activated by STIM1 following depletion of ER Ca²⁺ stores. A number of previous studies have been carried out over the last 10 years to try to understand how this process occurs, identifying various elements of the Orai channel and in its activator, STIM1, that are involved in CDI. How these various structural features support CDI has not been very clear, however.

Here, the authors have studied CDI by exploiting the fact that an alternative-initiation site variant of the Orai1 channel, Orai1-beta, largely lacks CDI. They have used electrophysiology and pharmacology, pull-down assays and FRET measurements to obtain results indicating that the Ca²⁺-activated adenylyl cyclase AC8 is associated with the N-terminal region (lacking in Orai1-beta) of Orai1; that AC8 is activated by Ca²⁺ influx through Orai1 to produce cAMP and thereby activate PKA localized to the complex by the A-kinase anchoring protein AKAP79; and that phosphorylation of a serine residue in the N terminal region of Orai1 mediates this channel's inactivation.

In addition, the authors provide results suggesting a possible role for regulated Orai1 CDI in cellular function: the naturally-occurring Orai1-beta variant produces more frequent oscillations in cytosolic Ca²⁺ in response to application of IP3R-pathway activating agonist (carbachol), and when the Orai1 variant is mutated at the N-terminal domain serine supporting CDI so that Orai1 CDI is prevented/reduced, the mutated Orai1 supports more frequent Ca²⁺ agonist-activated Ca²⁺ oscillations, similar to Orai1-beta. Significantly, the more frequent oscillations for Orai1-beta correlate with a greater initial rate and total amount of cytosol-to-nucleus translocation of the Orai/calmodulin/calcineurin-regulated transcription factor NFAT4. In contrast, no differential nuclear translocation was observed for NFAT1 translocation.

The experimental results supporting these conclusions are strong and are actually quite a bit more comprehensive than indicated above. The paper is clearly written. The findings are certainly important

because they open up new avenues for investigation, both in CDI and in understanding differential control of signaling to the genome.

We sincerely thank reviewer 1 for all these positive comments on our work.

Specific points:

A significant problem for the way the story is presented in the manuscript is that Orai1-beta does clearly exhibit significant CDI, albeit much less than does Orai1. Orai1-beta CDI is eliminated, as for Orai1, when extracellular Ca²⁺ is replaced by Ba²⁺. Thus there is more going on, in regard to CDI, than is presented in the manuscript. This should be clearly addressed in the writing.

In the wording of the manuscript, I would suggest toning down the idea that this is the first major step forward in understanding CDI of Orai channels (even though there is a case to be made that this is true). A great deal of work has been done previously, without a lot of forward advance. But based on the present manuscript, those findings will ultimately need to be incorporated into a more complete description of CDI, particularly downstream from the phosphorylation step identified here. It may also be that CDI of Orai channels is more complicated than presented here, and it could also be different in different cellular contexts (excitable versus non-excitable cells studied here).

Thank you. This point, which has been raised by the other reviewers, has been clearly addressed throughout the manuscript, including the results and discussion sections (see highlighted sections throughout the manuscript).

Activation of Orai1 by the STIM1 activation domain (SOAR, CAD) peptide is known to produce Orai1 currents that do not exhibit fast CDI: how could that intriguing finding be incorporated into the mechanism reported here?

Thank you. We have added a couple of speculative sentences in the discussion to address this point (see page 10 and 11).

Reviewer #2 (Remarks to the Author):

The manuscript confirms the previously reported interaction between adenylyl cyclase 8 (AC8) and the ORAI1 N terminus. It further establishes that there is a pathway in HEK293 cells leading from AC8, through cyclic AMP and protein kinase A (PKA) positioned by AKAP79, to phosphorylation of ORAI1 S34. The phosphorylation of S34 enhances fast Ca²⁺-dependent inactivation (CDI) of ORAI1 currents. Fast CDI correlates with fewer cytosolic Ca²⁺ transients in HEK293 cells stimulated with low concentrations of the agonist carbachol, and there is a corresponding difference in NFAT4 nuclear translocation between cells expressing ORAI1, which is subject to the AC8 regulatory pathway, and cells expressing ORAI1 β , which is not.

The authors tackle a difficult question, and make a compelling case that an AC8 > cAMP > PKA > S34 phosphorylation pathway is involved in regulating fast CDI in HEK293 cells. The experiments are

technically flawless, and the manuscript is very clearly written. An additional important finding is that fast CDI, usually measured under extreme conditions of Ca^{2+} influx, correlates with significant changes in cellular Ca^{2+} signaling under physiological conditions. The latter point deserves further exploration, though as part of a separate paper. Some questions of interpretation remain.

We are very grateful to reviewer 2 for the positive assessment of our work.

(1) Does the AC8 pathway account completely for fast CDI in HEK293 cells? Comparison of the currents recorded with BAPTA or with EGTA in the intracellular solution has been the defining measure of fast CDI triggered by Ca^{2+} binding to an intracellular site near the channel. Here, the degree of protection of ORAI1 current with BAPTA (74% remaining current for the -120 mV step, compared to 49% with EGTA) is much larger than for ORAI1 R31A-R33A, siAC8, H89, Ht-31 peptide, or ORAI1 S34A (where remaining current ranges from 53%-56%). The divergence suggests that another process might account for a large fraction of fast CDI in HEK293 cells.

This is a similar comment that has been raised by all reviewers. We agree that in addition to AC8 other mechanisms must account for a significant portion of ORAI1 CDI. We have clearly addressed this point throughout the manuscript, including the results and discussion sections (see highlighted sections throughout the manuscript).

(2) Are the AC8 pathway and phosphorylation of ORAI1 S34 required for fast CDI? If the question is taken in a literal sense, the answer is no. A considerable degree of fast inactivation is seen with ORAI1 S34A when Ca^{2+} is the current carrier (Figure 4e and 4i), and is absent when Ba^{2+} is the current carrier (Supplementary Figure 7b), satisfying a criterion for Ca^{2+} dependence. However, it might be that Ca^{2+} acts in this case at a site not accessible to BAPTA (Scrimgeour et al. 2012 Biochem. J. 441,743; Scrimgeour et al. 2014 Biochim. Biophys. Acta 1838, 1281), and, if so, the literal interpretation might include mechanisms outside the conventional notion of fast CDI. Rephrasing the question to ask whether a second intracellular site near the channel is linked to fast CDI, an experimental test would be whether BAPTA still has a protective effect in the case of ORAI1 S34A.

Per reviewer 2 suggestion, we have performed additional patch clamp recordings comparing CDI of ORAI-S34A in EGTA and BAPTA and show clearly that ORAI-S34A CDI is significantly reduced in BAPTA compared to EGTA (now depicted in Supplementary FIG. 8a-c), suggesting that a distinct intracellular Ca^{2+} binding site is linked to ORAI1 CDI and that phosphorylation of serine-34 likely contributes to CDI by recruiting more ORAI1 channels into an inactive state. Please see results section, page 7 with highlighted text.

(3) The manuscript does not offer evidence that it is Ca^{2+} entering through the ORAI1 channel that, within a few milliseconds, activates the AC8 pathway controlling fast CDI. A plausible alternative is that a pre-existing level of S34 phosphorylation sets ORAI1 susceptibility to a distinct fast CDI process. In fact there is background S34 phosphorylation in the absence of forskolin in Figure 4k. (It is difficult to assign functional relevance or irrelevance to the apparent level of background phosphorylation, since it has not been determined how many of the six S34 sites in a channel need to be phosphorylated to enhance fast CDI.) Forskolin increases S34 phosphorylation, as intended, and in fact all the tested conditions designed to manipulate the AC8 pathway could potentially change the pre-existing level of S34 phosphorylation.

We agree with the reviewer assessment. As stated in response to comment #2, additional Ca²⁺ binding site(s) are likely involved. We removed statements from the introduction implying that immediate Ca²⁺ entering through ORAI1 within ms range is what triggers the AC8-mediated effect on CDI.

(4) There is a further reason to be concerned about the direct connection between Ca²⁺ entering through the channel and the AC8 pathway. The turnover number reported for engineered soluble mammalian AC (30 s⁻¹, Dessauer and Gilman 1997 J. Biol. Chem. 272, 27787) seems too sluggish to drive a process whose τ_{fast} is 5 ms. Ideally the issue would be settled by detecting a rapid increase in S34 phosphorylation, under the relevant conditions for fast CDI, correlated with the amount of Ca²⁺ influx, and on the millisecond time scale. The ideal experiment would be prohibitively difficult at present, but the authors should at least note that they are proposing a specialized mechanism that might entail, for example, AC8 poised for a single enzymatic turnover and direct transfer of cyclic AMP to PKA tethered nearby.

Thank you. We agree with the reviewer's assessment that AC8 turnover is likely too slow to account for the trigger of CDI. This has been discussed in results section (see above response to comment #2).

(5) Statements in the title, abstract, and text that fast CDI has a role in shaping transcription are unsupported, since no data on transcription are presented in the manuscript.

Thank you. In title, abstract and throughout the text, we have changed the word "transcription" with "NFAT activation" to be more accurate.

(6) Discuss whether it is likely there are cell-type differences in fast CDI. HEK293 cells are of neuroepithelial origin, and AC8 expression is high in neuroepithelial-lineage cells. Is the same mechanism prominent in other cell types?

Thank you for this excellent suggestion. We have added a statement in the discussion to address this point. Please see highlighted text in page 11.

(7) The average numbers of Ca²⁺ oscillations in the example traces of Figure 5a, 5c, and 5e deviate substantially from the means plotted in Figure 5f, in a way that makes ORAI1 and ORAI1 S34D appear less active, and ORAI1 S34A more active. One way to obtain a more representative set of example traces would be to present four traces selected randomly from the lowest quintile for each condition, four traces from the next quintile, and so on.

Thank you. This excellent suggestion was implemented. Please see new Figure 5.

Reviewer #3 (Remarks to the Author):

Overall, the findings of Zhang et al, regarding the mechanism of Orail CDI are potentially interesting to the field. Their main finding is that Orail CDI is controlled, at least in part, by phosphorylation of the Orail N-terminal tail (likely at S34) by PKA. They propose that PKA is activated by cAMP generated from Ca²⁺-dependent adenylyl cyclase AC8 and recruited to the membrane by AKAP. Although interesting, the main problem with the study is that these effects are very small and the authors overstate their conclusions regarding the centrality of the PKA effect for inactivation. The AC8/PKA model is framed as THE sole mechanism for CDI but the results here do not show that. The conclusions of this

study will have to be significantly scaled down to be appropriate to the results shown.

We sincerely thank reviewer 3 for recognizing the potential importance of our work and are grateful for the constructive criticisms. We have listened to reviewer 3's criticism and agree with his/her assessment. Please see our response to reviewer 1 and 2.

1) The majority of the changes shown in CDI seen are in the 10-20% range (the key experiments in Figures 2-4 make this very clear). The Y-axis scales used for the plots and the high number of cells in each group (10-25 cells) make the effects appear highly statistically significant. However, the authors overstate the contribution of this pathway for CDI when there are clearly many other contributing factors.

Thank you. As stated in our response to reviewer 1 and 2, we have significantly revised the manuscript to reflect contributions from other pathways involved in ORAI1 CDI.

2) Orai1b still inactivates substantially even without the proposed phosphorylation site, none of the manipulations shown are able to recapitulate the entire difference in CDI between Orai1 and Orai1b, and the connection between their AC8/PKA/AKAP effects and S34 mutants appears to be tenuous. The authors need present this pathway as one contributor to CDI instead of THE mechanism.

Thank you. We agree with reviewer 3 and apologize for not clarifying this in the first version of the manuscript. As stated above, we have framed our findings within the published literature and have unequivocally stated that AC8 alone CANNOT account for the full extent of ORAI1 CDI (see highlighted text in page 4 and throughout the manuscript).

3) Could the changes in CDI arise from changes in STIM1-binding? STIM1 binding is essential for CDI (see relevant papers by Lewis, Rychkov) so any changes in the binding affinity of Orai1b channels lacking the N-terminus could influence the level of CDI. There is no data at all shown on this critical point. It is essential for this study to show that STIM1 binding is not altered in Orai1b as well as by manipulations (e.g. AC8/PKA pharmacology, S34 mutants).

We sincerely thank reviewer 3 for suggesting this excellent experiment. We performed FRET experiments to determine STIM1/ORAI1 and STIM1/ORAI1 β interactions under basal condition and under conditions of store depletion by thapsigargin and PKA inhibition by H89. These data are now included as Supplementary Fig. 10. The results of these experiments (which were repeated with several cells from independent transfections) were very surprising. We expected to see either no difference in FRET between ORAI1/STIM1 and ORAI1 β /STIM1 or a slight increase in ORAI1/STIM1 FRET that would be blocked by PKA inhibition, positioning the known role for STIM1 in ORAI1 CDI downstream of PKA. Instead, our data clearly showed that ORAI1 β /STIM1 FRET is significantly bigger than that of ORAI1/STIM1, reflecting a closer interaction between ORAI1 β and STIM1. Further, neither ORAI1/STIM1 nor ORAI1 β /STIM1 FRET were affected by PKA inhibition. This suggests that STIM1 binding to both ORAI isoforms is not altered by PKA inhibition and most importantly that a stronger STIM1 interaction cannot explain the enhanced ORAI1 CDI.

In light of the fact that both ORAI1 isoforms require STIM1 for activation and show SOCE with similar magnitudes on maximal activation by thapsigargin, the functional importance of the closer STIM1/ORAI1 β interactions would require substantial experimentation to fully comprehend; we plan to follow up on this interesting observation. We are indeed very grateful to reviewer 3 for suggesting this experiment but hope reviewer 3 understands that the extensive studies needed to clarify this observation are beyond the scope of the current manuscript.

4) If Orail CDI is modulated by phosphorylation of S34, why do mutations of S34 to D/E residues have no effect on CDI? Figure 4j shows that WT Orail is not phosphorylated at baseline without addition of forskolin, which suggests that a phosphomimetic mutation should enhance CDI. It would seem from this result that the effects at S34 on CDI are independent of the AC8/PKA pathway.

We thank reviewer 3 for this excellent comment. As eloquently articulated by reviewer 2 in his/her comment #3 and #4, it is very difficult to ascribe millisecond-scale role for S34 phosphorylation in ORAI1 fast CDI and AC8 turnover is likely too sluggish to fulfill the role of a trigger for fast CDI. As such, additional Ca²⁺-binding sites are likely involved. Therefore, under this scenario it is not surprising that the phosphomimetic mutants behave like wildtype ORAI1.

5) CDI is acutely sensitive to the local Ca²⁺ concentration around individual CRAC channels. Mullins et al, (JGP, 2016) have shown that some N-terminal Orail mutations elicit large changes in the single channel conductance of Orail channels. If this is also the case for the mutants and the Orailb isoform of this study, then amount of local calcium at the mouth of the channel would be very different in Orailb vs Orail channels (or the phosphorylation mutants). The authors would need to rule out this key possibility using noise analysis or other methods.

Reviewer 3 raises an interesting and quite valid criticism. Mutations within residues lining ORAI1 pore can indeed affect single channel conductance of ORAI1 (Mullins et al. JGP. 2016). However, it is highly unlikely that a mutation in serine-34, which is quite distal from the pore region, could alter single channel conductance. If this was the case, one would have to conclude that all manipulations performed within this study that are not aimed directly at ORAI1 or STIM1 (e.g. knockdown and overexpression of AC8, introduction of cAMP through the pipette, enhanced production of cAMP by Forskolin, expression of Δ I107-AKAP, inhibition of PKA, inhibition and knockdown of calcineurin) also alter ORAI1 single channel conductance, which is even more unlikely.

However unlikely this possibility might be, we strived to address reviewer 3's comment experimentally. As reviewer 3 knows, performing direct ORAI1 single channel measurements is simply impossible with current methods. Noise analysis is an indirect method, which relies on measurements of Na⁺ currents with a Ca²⁺ block protocol to compute single channel conductance from noise, something that our lab has never attempted before to have confidence in our abilities to perform it up to standards. Instead, we opted for an alternative experiment. We reasoned that by significantly lowering intracellular Ca²⁺ buffering (1 mM EGTA, instead of 10 mM) and enhancing driving force for Ca²⁺ (50 mM Ca²⁺ in bath instead of 20 mM), any potential difference between ORAI1, ORAI1 β and S34A-ORAI1 in single channel contribution to cytosolic Ca²⁺ would be blunted. These data represented in supplementary Fig. 9 show that even under these conditions differences in CDI persist between ORAI1, ORAI1 β and S34A-ORAI1.

6) If CDI is defective in Orai1b compared to Orai1 as concluded, why are the kinetics of inactivation faster as measured by tau 1 and tau 2? This does not make sense.

In light of this comment, we carefully re-analyzed our data to make sure that no mistakes have been made in the previous version. We confirmed that indeed, tau 1 and tau 2 are smaller for ORAI1 β . However, the time constants are not only a reflection of the kinetics but also of the proportion of current being inactivated. In other words, the taus of ORAI1 β inactivation also reflect the fact that ORAI1 β inactivates a lesser % of overall current. Please see figure 3 and 5 in Mullins et al. JGP 2016. 147:137-152; for a good example in particular please compare in Mullins et al. the taus of WT ORAI1 and ORAI1 W76K mutant in Figure 5.

7) In supplemental figure 8, a statistically significant difference in the “percentage of traces with one spike event” seems to be arbitrary, would be better to show histogram of the number of spike events in Orai1 vs. Orai1b cells. Also, how are “plateau events” defined? The authors will need to use more quantitative measures such as average duration or tau of decay of the events.

We thank reviewer 3 for this comment. In terms of regenerative Ca²⁺ oscillations, the one spike events are important: a cell showing 1-spike is unable to refill its ER to the threshold required for IP3 receptors to fire another spike. In fact, the lower the concentration of agonist, or the lower the driving force, or the amount of STIM1/ORAI1, the higher is the proportion of cells with a one-spike event. We emphasized that our comparisons of frequency of Ca²⁺ oscillations between ORAI1 variants include all oscillatory events, even cells displaying a one oscillatory event. Plateau events are now defined per reviewer 3 suggestion and example traces are shown for plateau, 1-spike event and regenerative oscillations in Supplementary Fig. 11. We found the durations and kinetics of oscillatory events to be similar across all conditions.

8) The discussion only alludes to the CaM model of CDI - the authors should also mention the proposed roles of the N-terminus (Frischauf 2011), TM2-3 loop (Srikanth 2010, Frischauf 2011), C terminus (Lee 2009), inner pore residues (Mullins 2016)...etc. Can their model explain the differences in CDI between Orai1/2/3 isoforms? How could the modulation of the N-terminus mediate CDI? It is clear that other regions also play roles in CDI, so their AC8/PKA model should be not be framed as the sole mechanism.

Thank you for this excellent suggestion, which was also noted by other reviewers (please see our response to reviewer 1 and 2). We have cited the references suggested by reviewer 3 and framed our findings in line with reviewer 3 suggestion that this is NOT the sole mechanism.

Minor typos:

· Second sentence of introduction is missing the word “current” at the end.

Thank you. This was added.

· Page 8, second paragraph line 8: should be “live-cell imaging”.

Thank you. This typo was fixed.

Reviewer #4 (Remarks to the Author):

The main finding is important, the study is detailed and control experiments are reasonable.

We are very grateful to reviewer 4 for recognizing the importance of our work.

One somewhat surprising finding is that the phenomenon involving Ca²⁺ entry, Ca²⁺-dependent cAMP production and PKA-dependent phosphorylation of ORAI1 is sufficiently fast to produce CDI (on time scale of few milliseconds). This could be perhaps discussed in conjunction with the known kinetics of cAMP production/PKA phosphorylation. In future (not necessarily for this paper) cAMP uncaging could be used to further characterise the surprisingly fast Ca²⁺-cAMP-PKA response.

Thank you for this comment. This issue has also been raised by reviewer 2. Please see response to reviewer 2, comments 2, 3 and 4.

It would be useful to test the sensitivity of NFAT4 nuclear translocation to cAMP-increasing agents (in cells expressing TK-ORAI, TK-ORAI1 β and one of S34 mutants)

We are very grateful to reviewer 4 for this outstanding suggestion. We would have liked very much to perform this experiment looking at NFAT4 translocation in intact live cells expressing either TK-ORAI, TK-ORAI1 β or TK-S34A ORAI1 mutant. Unfortunately, cAMP and PKA are known positive regulators of all three types of IP3 receptors that dramatically enhance IP3R-mediated Ca²⁺ release responses (see review by Taylor CW. Cell calcium, 2017. 63:48-52). For this reason, the effects of cAMP/PKA on IP3 receptors will confound the effects of TK-ORAI, TK-ORAI1 β or S34A mutant on NFAT nuclear translocation and hamper comparisons between these ORAI variants.

Minor point: P.6. Please specify “Online software tools...”

Thank you. This has been specified. Please see page 5, top.

Reviewers' Comments:

Reviewer #1:

Remarks to the Author:

The manuscript is substantially improved, satisfactorily addressing concerns in previous review.

However, for Fig. 5 and Supplemental Figure 10, were images really collected only every 10 min, as written in the Methods? If so, how were the E-FRET data in Supplemental Figure 10, presented at ~every 20 sec, obtained? If the rationale for infrequent imaging (every 10 min) was to avoid bleach, and if it turns out imaging was in fact carried out every 20 sec, how was the problem of bleach dealt with? Is the n value at EACH mean value plotted in panels a and b the same as indicated in panels b and d? This must be clarified, as the E-FRET time courses presented exhibit almost unbelievably low variance. These concerns regarding either describing or carrying out the FRET experiments deserve a very careful reply because they call into question whether this part of the manuscript is reliable.

Note that there are a number of typographical errors in the description of FRET in the Methods as well.

Reviewer #3:

Remarks to the Author:

I have no further comments.

Reviewer #4:

Remarks to the Author:

My main concern has been addressed by the authors (and the manuscript was modified accordingly).

The additional experiments recommended in my previous report are for the further development of this research area.

Reviewer 1:

The manuscript is substantially improved, satisfactorily addressing concerns in previous review.

However, for Fig. 5 and Supplemental Figure 10, were images really collected only every 10 min, as written in the Methods? If so, how were the E-FRET data in Supplemental Figure 10, presented at ~every 20 sec, obtained? If the rationale for infrequent imaging (every 10 min) was to avoid bleach, and if it turns out imaging was in fact carried out every 20 sec, how was the problem of bleach dealt with? Is the n value at EACH mean value plotted in panels a and b the same as indicated in panels b and d? This must be clarified, as the E-FRET time courses presented exhibit almost unbelievably low variance. These concerns regarding either describing or carrying out the FRET experiments deserve a very careful reply because they call into question whether this part of the manuscript is reliable.

Note that there are a number of typographical errors in the description of FRET in the Methods as well.

We are very grateful to reviewer 1 for spotting this discrepancy in the revised version of the manuscript. Our FRET methods reflect the experiments of AC8 and AKAP interactions with ORAI1 contained in the first version of the manuscript. We sincerely apologize for failing to update the methods section to reflect the additional FRET experiments between STIM1 and ORAI1 that were provided in the revised version of the manuscript. There are several key differences between these two different sets of experiments that we have now highlighted in the methods section and discussed below. We have also specified the fluorescent tag and its location (C- vs N-terminus) in Supplementary Fig. 10, changed the x-axis to “min” instead of “sec” to be consistent with other figures and corrected the few typos that reviewer 1 identified in the methods section. Our response to reviewer 1 comment is as follow:

For the new experiments represented in Supplemental Figure 10 (FRET between C-terminally tagged ORAI1/ORAI1 β and STIM1), images were collected every 20 seconds. In these studies, STIM1-YFP and ORAI1/1 β -CFP constructs demonstrate very strong expression, high fluorescence signal and minimal FRET at rest reflecting the two distinct locations (ER vs PM) of these proteins. Furthermore, the interaction between the C-termini of these proteins as measured by FRET in response to treatment with thapsigargin is robust, relatively fast (reaching maximal values within 5-7 min), making these constructs amenable to more frequent sampling over the relatively short 10 min time course. Conversely, the AC8/AKAP constructs demonstrate much lower expression compared to STIM/ORAI constructs with both AC8/AKAP and ORAI1 located in the PM. Further, interactions between N-terminally tagged AC8/AKAP and N-terminally tagged ORAI1 are constitutive and relatively weak, with no change in E-FRET between CFP-AC8/AKAP and YFP-ORAI1 in response to agonist stimulation. As such, to probe for potential long-term associations (up to 40 min), image collection was reduced to sampling every 10 minutes to avoid potential bleaching of the AC8/AKAP fluorescence.

The data provided in the revised manuscript as Supplemental Figure 10 is presented as mean +/- SEM at each time point for the total number of cells analyzed (n=73-100) originating from six independent coverslips for each of the four conditions shown. The scatter blots in “c” and “d” show the YFP/CFP ratio for each cell analyzed in “a” and “b”. Therefore, reviewer 1 is correct,

the n value at each mean value plotted in panels “a” and “b” is the same as indicated in panels “c” and “d” and this has been clarified in the legend to Supplementary Fig. 10. The variance for each data point during this time course is indeed very low (compared to experiments of AC8/AKAP interactions with ORAI1) reflecting the higher expression of these constructs, the quality of STIM1/ORAI1 interactions (consistently minor interactions at rest and robust interactions on store depletion), the magnitude of change of E-FRET between STIM1 and ORAI1 in response to thapsigargin, and to the higher sample size compared to AKAP/ORAI1 FRET experiments (n=73-100 vs 16-20 for AKAP/ORAI1).

Reviewers' Comments:

Reviewer #1:

None